# BACE1 regulates sleep–wake cycle through both enzymatic and non–enzymatic actions

Hannah Heininger[1,7], Xiao Feng [2,3,7], Alp Altunkaya[4,7], Fang Zheng [1], Florian Stockinger [1], Benedikt Wefers[2], Stephan A Müller [2,3], Pieter Giesbertz [2,3], Sarah K Tschirner [2,3], Dorina Shqau [2,3], Helmuth Adelsberger[5], Alexey Ponomarenko[1], Thomas Fenzl [4,8], Christian Alzheimer[1,8], Stefan F Lichtenthaler [2,3,6,8 ✉] & Tobias Huth [1,8 ✉]

## Abstract

**The β-secretase BACE1 has become a prime target in Alzheimer's disease (AD) therapy, because it drives the production of pathogenic amyloid β peptides. However, clinical trials with BACE1-targeting drugs were halted due to adverse effects on cognitive performance. We propose here that cognitive impairment by BACE1 inhibitors may be a corollary of a higher function of BACE1 related to proper sleep regulation. To address non-enzymatic effects of BACE1 on ion channels likely involved in the sleep-wake cycle, we analyze sleep patterns in both BACE1-KO mice and a newly generated transgenic line expressing a proteolysis-deficient BACE1 variant (BACE1-KI). We find that BACE1-KI and BACE1-KO mice display common and distinct sleep-wake disturbances. Compared with their respective wild-type littermates, both mutant lines sleep less during the light phase (when they preferentially rest). Furthermore, transition rates between wake and sleep states are altered, as are sleep spindles and EEG power spectra mainly in the gamma range. Thus, a better understanding of how BACE1 interferes with sleep-modulated behaviors is needed if clinical trials with BACE1-targeted inhibitors are to resume.**

**Keywords** BACE1; EEG; Sleep-Wake Cycle; Power Spectrum; Non-Enzymatic
**Subject Categories** Molecular Biology of Disease; Neuroscience

## Introduction

The β-secretase BACE1 (β-site amyloid precursor protein (APP) cleaving enzyme 1) plays a central role in Alzheimer's disease (AD), as it initiates the amyloidogenic processing of APP to form pathogenic amyloid β peptides (Aβ) (Hussain et al, 1999; Lin et al, 2000; Sinha et al, 1999; Vassar et al, 1999; Yan et al, 1999). So far, however, the inherent therapeutic implication that pharmacological inhibition of BACE1 should counteract AD (Luo et al, 2001) has not withstood clinical testing. In fact, trials were halted due to the occurrence of mild cognitive deterioration and other adverse effects (McDade et al, 2021). The disappointing outcome underscores the need to unravel the full spectrum of biological actions of BACE1.

Global and conditional BACE1 knockout (BACE1-KO) mice exhibit several neuropsychiatrically relevant phenotypes including increased locomotion and ataxia, spontaneous seizures, schizophrenia-related features, decreased thermal pain thresholds, and hearing loss, accompanied at the network and cellular level by neuronal hyperexcitability, aberrant synaptic transmission, axon guidance defects, and impaired axonal myelination (Kandalepas and Vassar, 2014; Weber et al, 2017). For most of these dysfunctions, it remains elusive which substrate is involved. It also remains unknown whether the proteolytic function of BACE1 is required or whether BACE1 may even have additional, non-proteolytic functions.

In fact, there is increasing evidence that some proteases—besides their proteolytic functions —may have additional, non-proteolytic functions. For example, presenilin 1 does not only act as a catalytic subunit of the protease γ-secretase but also functions as a calcium leak channel in the endoplasmic reticulum (Bezprozvanny, 2013). Another example is the metalloprotease MMP-12. After viral infection of cells, macrophages secrete MMP-12, which is taken up by epithelial cells, in which it acts as a transcriptional activator of IκBα expression, which in turn is required for efficient release of IFNα to fight viral infection (Marchant et al, 2014). Likewise, we showed previously that the catalytically silenced BACE1 D289N mutant is still able to regulate ion channels and neuronal excitability, most likely involving a direct protein–protein interaction with ion channel subunits (Agsten et al, 2015; Hartmann et al, 2018; Hessler et al, 2015; Huth et al, 2009).

[1]Institute of Physiology and Pathophysiology, Friedrich-Alexander-Universität Erlangen-Nürnberg, Erlangen, Germany. [2]German Center for Neurodegenerative Diseases (DZNE), Munich, Germany. [3]Neuroproteomics, School of Medicine and Health, TUM University Hospital, Technical University of Munich, Munich, Germany. [4]Department of Anesthesiology and Intensive Care, School of Medicine and Health, TUM University Hospital, Technical University of Munich, Munich, Germany. [5]Institute of Neuroscience, Technical University of Munich, Munich, Germany. [6]Munich Cluster for Systems Neurology (SyNergy), Munich, Germany. [7]These authors contributed equally: Hannah Heininger, Xiao Feng, Alp Altunkaya. [8]These authors jointly supervised this work: Thomas Fenzl, Christian Alzheimer, Stefan F Lichtenthaler, Tobias Huth. ✉E-mail: stefan.lichtenthaler@dzne.de; tobias.huth@fau.de

At present, it remains unclear whether non-proteolytic effects of BACE1 can also be seen in vivo. To test for such functions, we used CRISPR/Cas9 and generated mice with a mutation in the catalytic site of BACE1 (D289N), referred to as BACE1 knock-in (BACE1-KI) mice. In contrast to BACE1-KO mice, the BACE1-KI mice express BACE1, but the BACE1 protein lacks proteolytic activity. There is a high prevalence of disordered sleep in AD patients. Sleep deficits are also observed in patients treated with BACE1-targeted inhibitors (Egan et al, 2019; Naidu et al, 2025). Here, we used both mouse lines to examine the role of BACE1 in the sleep–wake cycle. Although sleep disturbances both drive and accompany neurodegenerative diseases, surprisingly little is known about the potential involvement of BACE1 in this fundamental neurobiological process. We found that BACE1 shapes sleep architecture and regulates sleep–wake transitions involving both proteolytic and non-proteolytic effects.

## Results and discussion

### Generation and validation of a proteolysis-deficient BACE1 knock-in (BACE1-KI) mouse

To test for non-proteolytic functions of BACE1, we generated a BACE1 knock-in (BACE1-KI) mouse line, in which one of the catalytic aspartic acid residues in the endogenous mouse BACE1 is mutated to asparagine (D289N, Fig. 1A). Mutant mice were identified by genotype analysis (Fig. 1B).

To prove that the D289N mutation indeed abrogated the enzymatic activity of BACE1, we generated membrane and soluble fractions of mouse brains, which contain the full-length membrane proteins and the BACE1-cleaved soluble substrate ectodomains, respectively (Kuhn et al, 2012; Pigoni et al, 2016; Zhou et al, 2012). We analyzed brains from two distinct cohorts of mice. From the BACE1-KI cohort, we compared wild-type (WT), heterozygous (HET) and BACE1-KI mice. As a control, we analyzed BACE1-KO mice (Cai et al, 2001) and the corresponding littermate control WT mice.

Seizure protein 6 (SEZ6) is a known BACE1 substrate (Kuhn et al, 2012; Pigoni et al, 2016). Its cleaved ectodomain (sSEZ6) was present in WT and HET BACE1-KI mouse brains but absent in BACE1-KI and -KO mice (Fig. 1C; Appendix Fig. S1), indicating loss of BACE1 proteolytic activity in the BACE1-KI line. Similar patterns were seen for sAPPβ, the BACE1-cleavage fragment of APP (Fig. 1D). In contrast, sNrCAM, cleaved by ADAM10, remained unchanged across all genotypes (Fig. 1C). Full-length APP, SEZ6, and NrCAM levels were unchanged in the membrane fraction. BACE1 protein was absent in BACE1-KO mice but unchanged in BACE1-KI mice (Fig. 1C,E). These results show that the BACE1 D289N mutant is present but proteolytically inactive.

BACE1 cleaves many neuronal substrates beyond SEZ6 and APP. To determine if the BACE1-KI mutant affects only APP and SEZ6, we used mass spectrometry to broadly assess BACE1 substrate processing in primary neurons from WT, HET, and BACE1-KI mice. Using the hiSPECS method (Tüshaus et al, 2020), we identified soluble ectodomains of BACE1 substrates, along with full-length proteins from neuronal lysates (Appendix Fig. S2 for workflow, Appendix Fig. S3 for data).

BACE1-KI neurons showed a marked reduction in BACE1-cleaved ectodomains, including known BACE1 substrates SEZ6, APLP1, CHL1, L1CAM, CACHD1, and others, compared to WT (Fig. 1F; Appendix Fig. S3) (Dislich et al, 2015; Hemming et al, 2009; Kuhn et al, 2012; Müller et al, 2023; Pigoni et al, 2016; Stützer et al, 2013; Voytyuk et al, 2018; Zhou et al, 2012). The effect size of the reductions for each protein was dependent on the extent to which it is cleaved by BACE1 and is consistent with similar data using the BACE inhibitor C3 (Tüshaus et al, 2020). Full-length substrates were unchanged or increased, consistent with reduced cleavage (Fig. 1G; Appendix Fig. S3). Similar trends were seen in BACE1-KI vs. HET neurons, though some changes did not reach significance after FDR correction due to higher variability in the samples (Fig. 1H,I). No significant differences were found between HET and WT neurons (Fig. 1J,K), indicating that partial BACE1 loss has minimal impact. The tryptic peptides identified by mass spectrometry for the secreted BACE1 substrate ectodomains exclusively mapped to extracellular domains of the substrates, verifying that the secretome reflects true cleavage products rather than full-length proteins released from damaged cells (Appendix Fig. S4).

Taken together, these in vitro and in vivo experiments demonstrate that BACE1-KI mice lack proteolytic BACE1 activity.

### Comparison between BACE1-KI and BACE1-KO mice regarding nerve fiber pathologies, M-current and motor functions

To demonstrate the broad versatility of BACE1-KI mice, we first examined their phenotypes in paradigms, in which BACE1-KO mice had already exhibited deficits that were thought to be of either proteolysis-dependent or -independent origin. We first looked at the axonal guidance defect in the infrapyramidal bundle of the hippocampus from BACE1-KO mice, which has been attributed to impaired cleavage of the cell adhesion molecule CHL1 (close homolog of L1) (Hitt et al, 2012; Ou-Yang et al, 2018). Consistent with the respective findings from BACE1-KO mice, the infrapyramidal bundle in BACE1-KI mice was significantly shortened and appeared disrupted in comparison to WT mice (Fig. 2A,D). In peripheral nerve fibers, hypomyelination is a hallmark of BACE1-deficiency (Dierich et al, 2019; Willem et al, 2006; Weber et al, 2017; Hu et al, 2006). In our experiments, impaired ensheathing was a prominent finding in sciatic nerve transections from BACE1-KI mice (Fig. 2B), demonstrating that the defect resulted from the loss of the enzymatic activity of BACE1. Similarly, inner ear nerve fibers originating from the spiral cochlear ganglion of BACE1-KI mice were as affected by hypomyelination as those from BACE1-KO mice in our previous study (Fig. 2C) (Dierich et al, 2019). Furthermore, two conspicuous features previously described in BACE1-KO mice (Dierich et al, 2019) were also observed in BACE1-KI mice: MBP-positive bulbous expansions along the path of the axons, and disarranged NF200-positive fiber connections around the synaptic layer, where the fibers are generally unmyelinated in wild-type mice, also exhibiting pronounced swellings. In summary, nerve fibers from BACE1-KO and BACE1-KI mice exhibited a very similar extent of hypomyelination, thus corroborating the notion that proteolysis-competent BACE1 is required for normal myelination.

 

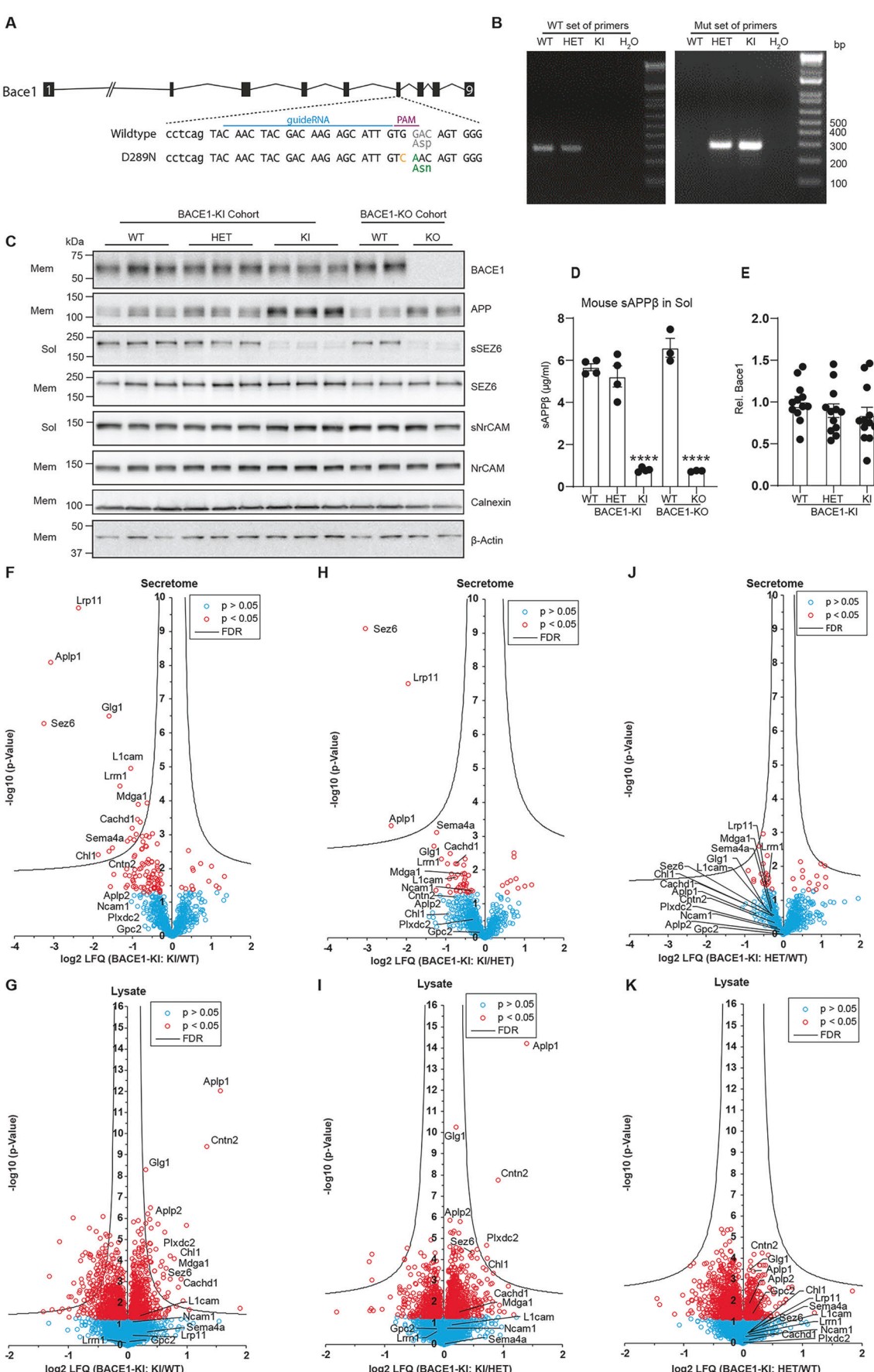

**Figure 1. BACE1-KI (D289N) mice show similar defective proteolytic activity to BACE1-KO mice.**

(A) Schematic representation of BACE1-KI was induced with a CRISPR-Cas9-mediated knock-in strategy. (B) Genotyping analysis of BACE1-KI mice with WT and mutant sets of primers. (C) Immunoblots of soluble (sol) and membrane (mem) fractions in BACE1-KI and BACE1-KO (Bace1$^{tm1Pcw/J}$) mice. BACE1, SEZ6, and APP were detected. NrCAM, Calnexin and β-actin served as loading control ($n = 12$). (D) ELISA analysis of sAPPβ in the soluble fraction of BACE1-KI and BACE1-KO mice. Data were shown as mean ± SEM ($n = 3$–4 for all groups). BACE1-KI and BACE1-KO conditions were compared against the respective wild-type (WT) control by one-way ANOVA with Tukey's multiple comparison test. Significant differences are indicated (****$p < 0.0001$). (E) Quantification of BACE1 in the membrane fraction from BACE1-KI mice. Data were shown as mean ± SEM ($n = 12$), one-way ANOVA with Tukey's multiple comparison test. (F–K) Volcano plots showing changes in protein levels in the secretome (F, H, J) and lysate (G, I, K) of primary cultured neurons from BACE1-KI mice using label-free quantitative proteomics. The log2-transformed LFQ intensity ratios are plotted against the −log10 transformed $p$ value of the Student's $t$-test. (red: proteins with $p < 0.05$; hyperbolic curve: permutation-based FDR significance threshold, $n = 6$–8). (F, G) BACE1-KI vs. wild-type (WT), (H, I) BACE1-KI vs. BACE1-KI heterozygous (HET) (J, K) BACE1-KI heterozygous (HET) vs. wild-type (WT).

By contrast, the two mouse lines differed significantly in the KCNQ-mediated M-current of hippocampal neurons. We used the depolarization of the membrane potential by the KCNQ-channel blocker XE991 (10 μM) as a proxy for the size of the standing M-current around resting potential. Irrespective of whether the neurons were pharmacologically isolated from synaptic input or not, XE991 produced a robust depolarization of about equal size in WT and BACE1-KI, whereas the drug effect was significantly smaller or virtually absent in BACE1-KO neurons (Fig. 2E,F). This finding strongly supports our concept that a direct non-enzymatic interaction between BACE1 and KCNQ-channel subunits is a necessary constituent of M-current functionality (Hessler et al, 2015).

We next assessed motor functions (Fig. 2G–J), replicating a previous study, in which BACE1-KO mice exhibited a mild-to-moderate ataxia-like motor phenotype, which became evident at high levels of difficulty, such as walking on a small elevated circular beam (Lehnert et al, 2016). As before, BACE1-KO mice performed poorly on the circular beam, whereas the relative slip frequency of BACE1-KI mice in this task ranged between that of BACE1-KO and that of WT mice (Fig. 2J). Given that the neuropathological aberrations implicated in motor deficits of BACE1-KO mice such as hypomyelination (Dierich et al, 2019; Willem et al, 2006; Weber et al, 2017) and paucity of muscle spindles and Pacinian-type mechanoreceptors (Cheret et al, 2013; Fleming et al, 2016) have all been attributed to the abrogated cleavage of NRG1 by BACE1, we consider the rather small difference in motor performance between the two transgenic BACE1 lines to be insufficient to establish a robust non-proteolytic contribution of BACE1.

## BACE1-KI and BACE1-KO mice display common and distinct sleep–wake disturbances

To address the still unresolved question of whether BACE1 is involved in the regulation of the sleep–wake cycle, we performed 23 h-long EEG recordings in freely moving BACE1-KO and BACE1-KI mice, and their respective littermate controls. During these recording sessions, we never observed electrographic seizures, in agreement with previous reports that such events occur only very infrequently (Hitt et al, 2010; Hu et al, 2010). In the compressed EEG recordings of Fig. 3A,B, each covering an entire 12 h light or 11 h dark phase, non-rapid eye movement sleep (NREMS) epochs are indicated in black, whereas wakefulness (WAKE) and rapid eye movement sleep (REMS) epochs are given in red and blue color for BACE1-KO and BACE1-KI mice and their control littermates in gray color, respectively. When EEG segments from all three vigilance states were plotted on a semi-diurnal time scale, both BACE1-KO and BACE1-KI mice showed an increase in EEG amplitude during WAKE states compared to their controls

(Fig. 3C,D, quantified in Fig. 5). NREMS-associated sleep spindles were detected in all groups (Fig. 3E,F), but upon closer examination, their features displayed appreciable variations between genotypes (see below).

Statistical analysis showed that mice from the two transgenic lines spent significantly more time in the WAKE state during the light phase than their respective WT littermates. This difference did not hold during the dark phase, when the mice, being nocturnal animals, are much more active. (Fig. 4A,D). In BACE1-KO mice, prolongation of WAKE states in the light phase occurred at the expense of both NREMS and REMS (Fig. 4A), whereas in BACE1-KI mice, only the total duration of REMS in the light phase was reduced (Fig. 4D). In the light phase, BACE1-KI mice exhibited more WAKE ▸ NREMS and fewer NREMS ▸ REMS and REMS ▸ NREMS transitions than their WT counterparts, whereas in BACE1-KO, the rate of transitions between the states in the light phase did not significantly differ from control (Fig. 4B,E). Conversely, the average bout duration of WAKE and REMS states during the light phase was longer in BACE1-KO mice than in control littermates, whereas no significant difference was found between BACE1-KI and control mice (Fig. 4C,F). Analysis of sleep spindles revealed an increased density during the light phase as a common abnormality in both transgenic lines (Fig. 4G,H), whereas in the dark phase, only BACE1-KO mice showed this feature. In contrast, higher amounts of spindles and lower spindle amplitudes were abnormalities during the light phase that were restricted to BACE1-KI mice and BACE1-KO mice, respectively (Fig. 4G,H).

The divergent EEG amplitudes between the different mouse lines prompted us to calculate their power spectra. An intriguingly consistent and dark-light cycle-phase-independent phenotype emerged across all vigilance states (Fig. 5A,B). The power spectrum was increased across all spectral bands. The effect was markedly pronounced at higher spectral bands (η, β, γ).

In view of our previous findings, an obvious question is whether the BACE1-mediated enhancement of M-current is involved in sleep regulation. Lending strong support to this idea, Li et al reported that a selective downregulation of KCNQ2/3 channels in arousal-promoting orexin neurons of the hypothalamus leads to sleep instability, whereas pharmacological augmentation of M-current promotes sleep quality (Li et al, 2022). In addition, lack of the KCNQ4-mediated M-current in cholinergic neurons of the pedunculopontine nucleus, which are part of the reticular activating system, was found to impair the ability of mice to adapt their sleep–wake cycles to changes in light and dark phases (Bayasgalan et al, 2021). Notwithstanding presumable effects of altered KCNQ-mediated currents on sleep cycle regulation, the behavioral and electroencephalographic differences between BACE1-KO and

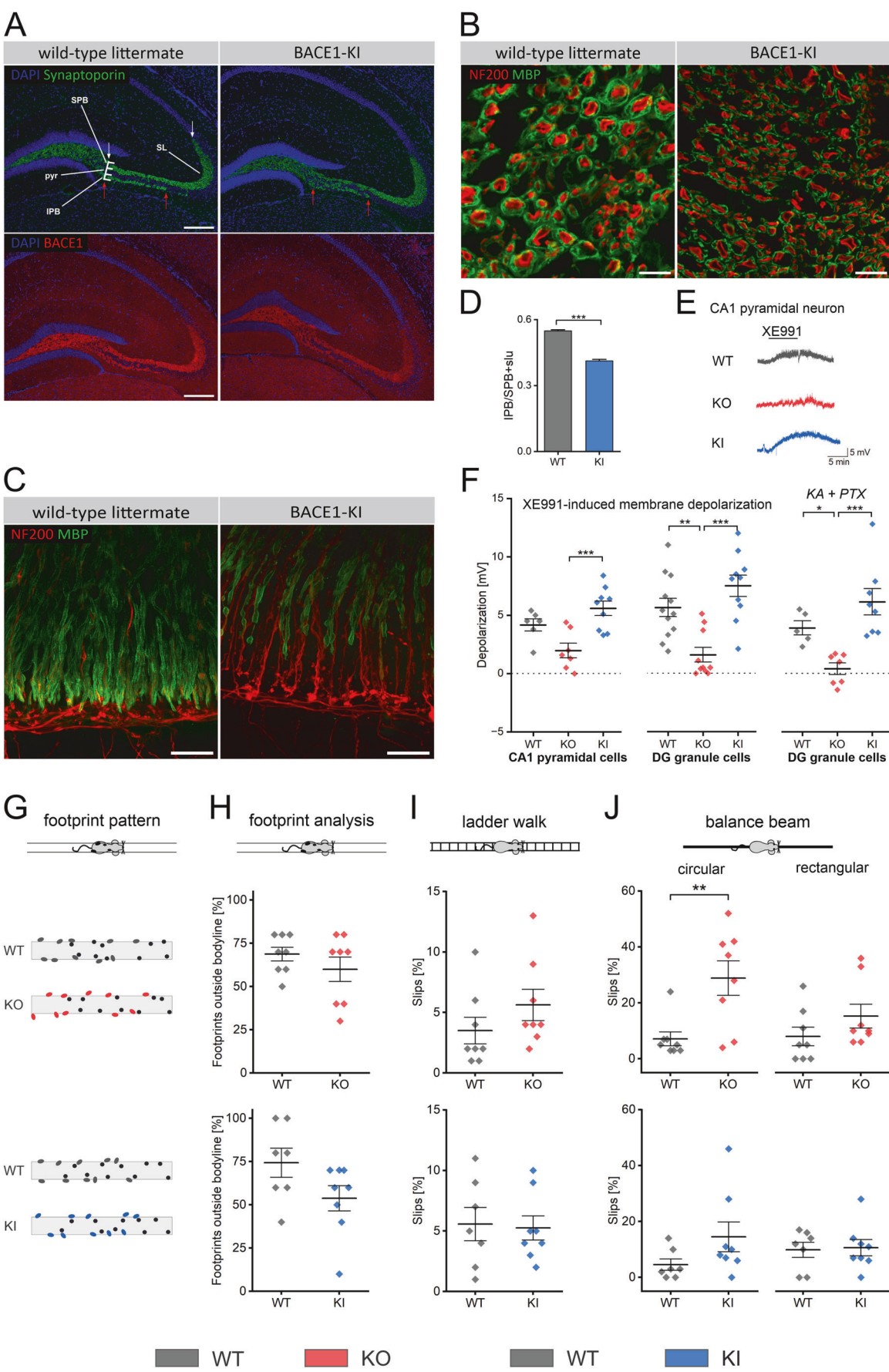

◄ **Figure 2. Established BACE1-KO phenotypes revisited using the BACE1-KI mouse model.**

(A–D) BACE1-KI show altered configuration of central and peripheral nerve fibers. (A) Hippocampal sections from the dentate gyrus/CA3 region of both BACE1-KI and wild-type (WT) littermates were immunostained for synaptoporin (top panels) and BACE1 (bottom panels). Synaptoporin labeling highlights the infrapyramidal (IFB) and suprapyramidal (SFB) fiber bundles, located in the *stratum lucidum* (SL) and adjacent to the *stratum pyramidale* (PYR). The lower panels show corresponding BACE1 staining. Scale bar = 200 μm. (B, C) Peripheral axons of BACE1-KI mice exhibit hypomyelination. Nerves were labeled with antibodies against Myelin Basic Protein (MBP) to visualize myelin sheaths and Neurofilament 200 (NF200) to identify axonal cores. (B) Compared to WT nerves (left panel), transverse sections of the sciatic nerve in BACE1-KI mice (right panel) show significantly reduced myelination. Scale bar = 10 μm, (*n* = 3). (C) A similar phenomenon is observed in type I cochlear nerve fibers innervating the inner hair cells. In WT mice (left panel), afferent fibers initially lack myelin but become uniformly myelinated further along, with remaining NF200-positive fibers representing primarily unmyelinated efferents. In contrast, BACE1-KI mice (right panel) display discontinuous, patchy myelination, more prominent NF200 staining due to partially unmyelinated afferent fibers, and swollen, disorganized axonal terminals. Scale bar = 20 μm, (*n* = 4). (D) The lengths of the IFB (red arrows in 2A) and SFB (white arrows in 2A) were quantified, and their ratio was calculated. Data were presented as mean ± SEM (*n* = 5 BACE1-WT; *n* = 4 BACE1-KI, mean of *n* ≥ 5 slices for each animal), ***$p < 0.001$. Statistical significance was determined using a Student's *t*-test. (E, F) M-currents were not altered in hippocampal neurons of BACE1-KI mice. Patch-clamp recordings were performed under whole-cell current-clamp mode in adult hippocampal slices. (E) Representative recordings from hippocampal cells in WT, BACE1-KI and BACE1-KO slices show the reversible responses to XE991 (10 μM). (F) Summary of the XE991-induced membrane depolarization. CA1 pyramidal cells and DG granule cells had comparable resting membrane potentials among these groups (CA1 pyramidal cells: WT −76.3 ± 2.0 mV; BACE1-KI −75.3 ± 0.9 mV, BACE1-KO −76.6 ± 0.8 mV; DG granule cells: WT −86.3 ± 0.7 mV; BACE1-KI −85.2 ± 1.2 mV, BACE1-KO −85.0 ± 1.2 mV, mean ± SEM). In some experiments, DG granule cells were pharmacologically isolated from GABAergic and glutamatergic synaptic inputs by Picrotoxin (PTX, 100 μM) and kynurenic acid (KA, 2 mM), respectively. Statistical comparisons were performed using one-way ANOVA followed by Tukey's post hoc test or two-tailed paired *t*-test. (*$p < 0.05$; **$p < 0.01$; ***$p < 0.001$). Data were presented as mean ± SEM, left panel: WT *n* = 6, KO *n* = 7, KI *n* = 9, middle panel: WT *n* = 12, KO *n* = 10, KI *n* = 10, right panel: WT *n* = 5, KO *n* = 7, KI *n* = 8. (G–J) Motor function is not significantly impaired in BACE1-KI mice. BACE1-KO mice, BACE1-KI mice, and their corresponding WT littermates underwent a series of motor coordination tests to assess ataxic behavior (see methods). (G) Mice were filmed from below as they walked across a transparent plate. Footprint patterns were analyzed to determine the positions of the paws. Front paw positions are shown as black circles, and hindpaw positions are color-coded. (H) The frequency of hindpaw placements occurring outside the animal's body line was quantified for statistical analysis. Additional tests included climbing a horizontal ladder (I), and traversing both a circular beam (J, left panel) and a rectangular beam (J, right panel), with the number of hindpaw slips recorded. Data were presented as mean ± SEM. Sample sizes: BACE1-KO (*n* = 8) vs. WT littermates (*n* = 8); BACE1-KI (*n* = 8) vs. WT littermates (*n* = 7). Statistical analyses were performed using a Student's *t*-test (**$p < 0.01$).

BACE1-WT were rather subtle. In contrast, the two transgenic mouse lines exhibited a number of common aberrations from the physiological sleep–wake cycles suggesting that it is predominantly the loss of enzymatic activity of BACE1 that defines the circadian phenotype of the mutant mice.

In this regard, identification of the substrates whose cleavage bears the strongest significance for the regulation of the sleep–wake cycle remains a daunting task for future studies. Notwithstanding these open questions, our study firmly establishes BACE1 as a molecular player that consolidates sleep stability in healthy adult mice. The picture becomes obfuscated, however, when we consider BACE1 in the context of the sleep pathologies typically seen in Alzheimer patients. Does BACE1 propel the disintegration of sleep architecture in AD, or does BACE1, e.g., by counteracting the age-associated decline in the M-current of orexin neurons (Li et al, 2022), make a (futile) attempt to slow sleep fragmentation?

Evidence for the former view comes from a study with the 5xFAD mouse model of Alzheimer's disease (Yao et al, 2023). Aberrant sleep organization in these mice is characterized by increased wakefulness at the expense of REMS and NREMS, recapitulating the disrupted sleep architecture of Alzheimer patients. Importantly, both conditional *Bace1* deletion and administration of the BACE1 inhibitor lanabecestat reversed the sleep disturbances in the 5xFAD mouse model. This finding is in contrast to results from a randomized clinical trial with the BACE1 inhibitor verubecestat in mild-to-moderate AD (Egan et al, 2019). According to this study, about 10% of patients receiving the drug (and about 5% of patients from the placebo group) reported sleep disturbance as an adverse event that occurred within 14 days after the last dose over the course of 78 weeks. Importantly, sleep disturbances have also been observed in clinical trials with other BACE1 inhibitors (Naidu et al, 2025), suggesting that this adverse effect is not an off-target action attributable to a particular drug, but rather results from the suppression of a physiological function of BACE1.

In addition to its presumed effects on the circuits of the brainstem and hypothalamus that regulate sleep and produce wakefulness, our EEG recordings implicate BACE1 in the modulation of the thalamocortical networks whose interplay generates sleep spindles in NREMS states. Irrespective of whether NREMS during light phase was reduced (BACE1-KO mice) or remained unchanged (BACE1-KI mice), both transgenic lines exhibited a significantly higher density of sleep spindles than their control littermates. Thus, it seems unlikely that spindle activity increases primarily to compensate for an overall reduction in NREMS. Rather, BACE1 may directly target one or more of the players in the thalamocortical loop that give rise to sleep spindles. While the exact mechanisms by which BACE1 alters spindle generation during NREMS awaits further study, the finding by itself should be of translational relevance. Sleep spindles are widely recognized as a central mechanism for memory consolidation, and patients with mild cognitive impairment (MCI) and AD have fewer sleep spindles and lower spindle activity than age-matched controls (Mander, 2020). We are not aware of any published polysomnography studies conducted during clinical trials of BACE1 inhibitors in AD patients. Thus, it remains to be determined whether our prediction from the sleep spindle dataset, namely a rescue of reduced spindle activity in AD patients by BACE1 inhibitors, withstands clinical examination.

With respect to the impact of BACE1 on the power of the different frequency bands in the EEG, a consistent finding in both transgenic lines was a substantial increase mainly in the gamma range. This effect was prominent in all sleep states. Network oscillations with gamma rhythmicity are widely considered as a means to synchronize cortical information processing during high-level cognitive functions, complex motor tasks, and deep emotional experiences. In a sharply contrasting view, however, Ray and Maunsell argued that gamma oscillations

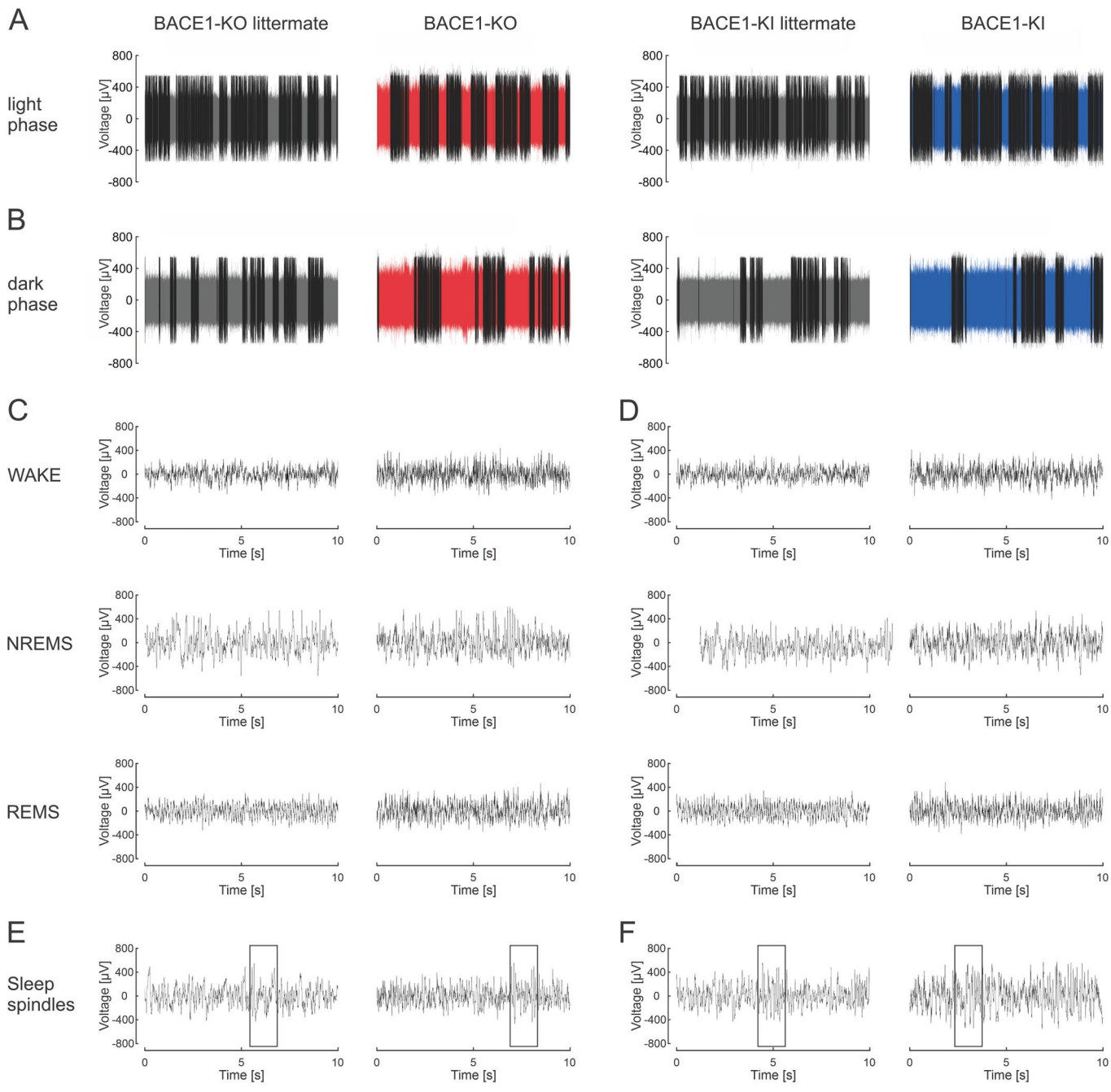

**Figure 3. Intracranial EEG recordings in freely moving BACE1-KO and BACE1-KI mice, as well as their littermate controls, in a 12 h light/12 h dark cycle.**

(A, B) Representative sleep–wake transitions are shown for a 12 h light period followed by an 11-h dark period. (C, D) Recordings were semi-automatically classified into wakefulness (WAKE), non-REM sleep (NREMS), and REM sleep (REMS) states (see Methods), and typical episodes are depicted. (E, F) Sleep spindles were identified (see Methods), with representative examples illustrated (boxed regions). (A–D) Note that both BACE1-KO and BACE1-KI mice exhibit similarly increased EEG amplitudes.

are unlikely to serve higher brain functions but rather represent a low-level signature of cortical interactions involving excitation and inhibition (Ray and Maunsell, 2015). Adopting such a reductionist concept, we may explain the relative increase in gamma rhythm in BACE1-KO and BACE1-KI mice in terms of altered features of intrinsic and synaptic inhibition, based on our current knowledge of how BACE1 affects brain neurophysiology. At the cellular level, a decrease in M-current, as measured in

BACE1-deficient mice (see above), has been linked to augmented gamma power (Klemz et al, 2021). Since an equal enhancement of gamma rhythm was seen in the two transgenic lines, which differ with respect to the size of their M-current, further effects of BACE1 should play a role. A likely candidate is the dual effect of the secretase on GABAergic inhibition. Previous work has shown that BACE1 reduces the surface expression of Nav1.1 (Kim et al, 2007) which is also diminished in a hAPP transgenic

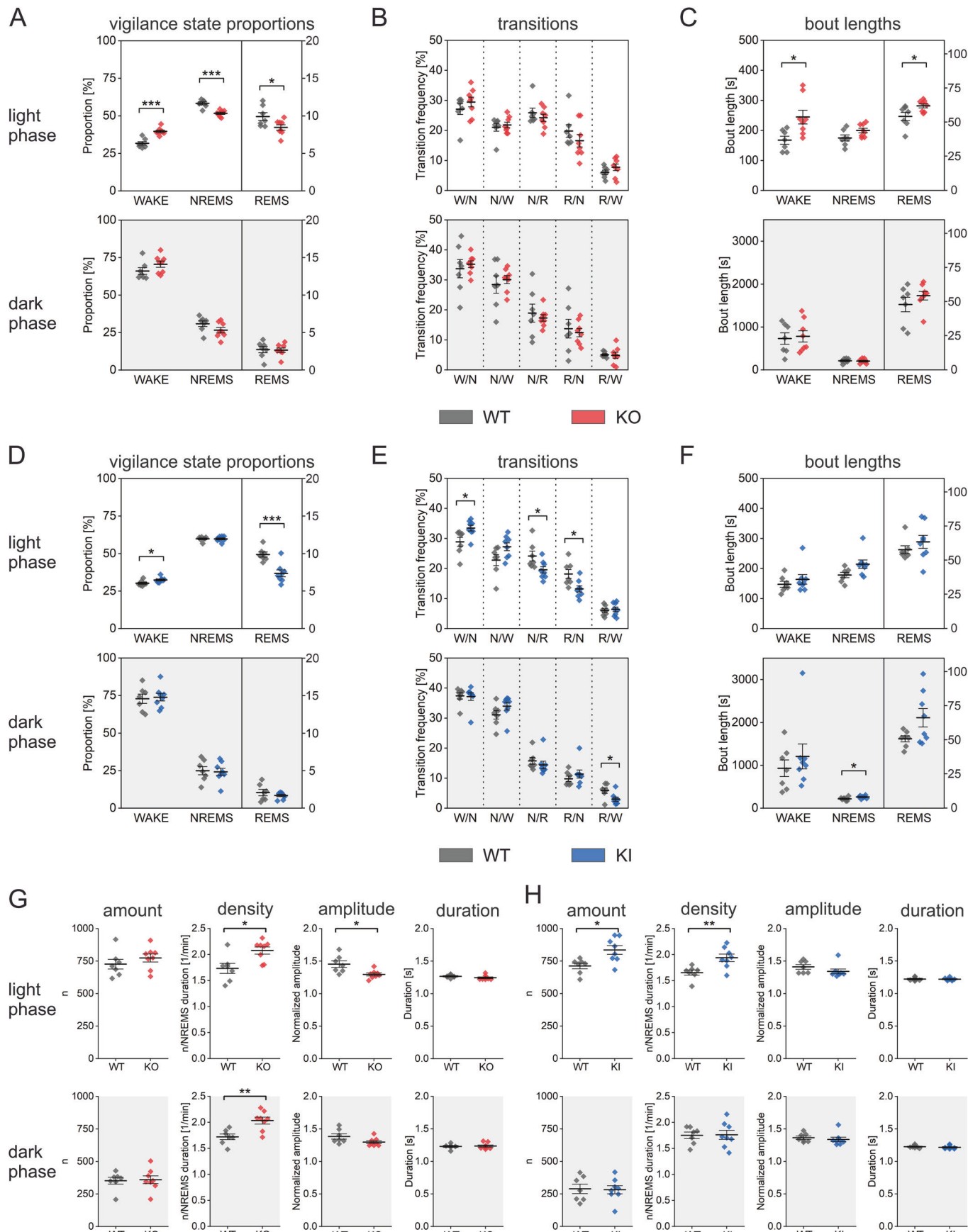

◄ **Figure 4. Transgenic BACE1-KO and BACE1-KI mice display altered sleep behavior compared to their corresponding littermate controls.**

EEG recordings were obtained from freely moving animals with a 12 h light phase and a 12 h dark phase (11 h recorded). Sleep stages were classified based on EEG spectral patterns and concurrent EMG activity, distinguishing between WAKE (W), non-REM sleep (NREMS, N), and REM sleep (REMS, R). Data were analyzed separately for the dark and light phases. (A, D) show the overall proportions of each sleep stage, (B, E) depict the frequency of transitions between these stages, and (C, F) illustrate the average bout length of each stage. Panels (G, H) summarize measures of sleep spindle activity during NREMS, including the total number, density, amplitude, and duration of spindles. (A–H) Data were presented as mean ± SEM. Sample sizes: BACE1-KO ($n = 8$) vs. WT littermates ($n = 7$), and BACE1-KI ($n = 8$) vs. WT littermates ($n = 7$). Statistical analyses were performed using a Student's $t$-test (*$p < 0.05$; **$p < 0.01$; ***$p < 0.001$).

mice model and AD patients (Verret et al, 2012). This subtype of voltage-gated $Na^+$ channels is essential to drive action potential discharges of GABAergic interneurons. In addition to hobbling interneuron firing, a rise in BACE1 activity was found to impair GABAergic inhibition by cleaving $GABA_A$ receptors, which serve as major effectors of synaptic inhibition in the postsynaptic neuron (Bi et al, 2025). Conversely, loss of BACE1 protein would be expected to strengthen and sharpen GABAergic inhibition. Since the generation of gamma oscillations critically depends on strong and precisely timed $GABA_A$ receptor-mediated IPSCs, we would argue that, given the apparently inverse relationship between BACE1 and GABAergic inhibition, the lack of BACE1 is likely to foster this mechanism, thereby promoting gamma rhythm.

In conclusion, our EEG recordings show that BACE1 modulates the power spectrum of cortical frequency bands and contributes to the maintenance of regular sleep–wake cycles in adult mice. It is conceivable that the overall impact of BACE1 on the manifestation of vigilance states within the circadian rhythm changes during normal aging and even more so in neurodegenerative diseases, as the putative targets of BACE1 may undergo substantial changes. As we used germline KO and KI mice, we cannot make firm predictions on how a treatment regimen with BACE1 inhibitors beginning much later in life may affect sleep parameters, all the more so as patients are likely to be afflicted by disease-related sleep disorders. Despite the limited translational perspective of our study, our findings suggest that if clinical trials of BACE1-targeting inhibitors are to be resumed, they should include polysomnographic recordings to assess whether the drugs are interfering with normal sleep or, more likely in AD patients, exacerbating preexisting sleep disturbances, thereby further compromising the many beneficial effects of sleep on cognitive performance, motor learning, and emotional stability.

## Methods

### Reagents and tools table

| Reagent/resource | Reference or source | Identifier or Catalog Number |
|---|---|---|
| **Experimental models** | | |
| BACE1em2Bwef (Bace1 D289N knock-in, "BACE1-KI", KI) | This study (CRISPR/Cas9) | BACE1em2Bwef |
| BACE1tm1Psa (Bace1 knockout, "BACE1-KO", KO) | Dominguez et al, 2005 | BACE1tm1Psa |
| B6.129-Bace1tm1Pcw/J (Bace1-KO) | Jackson Laboratory | Strain B6.129-Bace1tm1Pcw/J |
| C57BL/6 J | Charles River | - |
| **Recombinant DNA** | | |
| ssODN donor "ssBace1-D289N" for CRISPR HDR | This study | - |
| Genotyping primers (WT) | This study | - |
| Genotyping primers (Mutant) | This study | - |
| **Antibodies** | | |
| Anti-BACE1 (D10E5) | Cell Signaling Technology | #5606 |
| Anti-APP (22C11) | Merck Millipore | MAB348 |
| Anti-SEZ6 (soluble ectodomain) | Gift from Dr. Jenny Gunnersen | - |
| Anti-NrCAM | Abcam | ab24344 |
| Anti-β-actin | Sigma-Aldrich | A5316 |
| Anti-Calnexin | Enzo Life Sciences | ADI-SPA-860 |
| HRP-conjugated anti-mouse IgG | Dako | P0447 |
| HRP-conjugated anti-rabbit IgG | Dako | P0448 |
| Rabbit anti-NF200 | Sigma-Aldrich | N4142 |

| Reagent/resource | Reference or source | Identifier or Catalog Number |
|---|---|---|
| Mouse anti-MBP | Santa Cruz Biotechnology | sc-271524 |
| Goat anti-rabbit Cy3 | Jackson ImmunoResearch | 111-165-144 |
| Goat anti-mouse Alexa Fluor 488 | Molecular Probes | A-11029 |
| Rabbit anti-synaptoporin | SynapticSystems | 102 002 |
| Donkey anti-rabbit Alexa Fluor 488 | Molecular Probes | A-21206 |
| **Oligonucleotides and other sequence-based reagents** | | |
| **Oligonucleotides for generation of Bace1(D289N) mice** | | |
| **Protospacer + PAM** | | |
| crBace1_D289N | CAACTACGACAAGAGCATTGTGG | |
| **Mutagenic ssODN - Sequence (5′-3′)** | | |
| ssBace1-D289N | GATTCACCCCTCCCACCTCTCCCCCCCTCTCTCCTCAGTACAACTACGA CAAGAGCATTGTCAACAGTGGGACCACCAACCTTCGCTTGCCCAAGAA AGT ATTTGAAGCTGCCGTCAAGTCC | |
| **Genotyping and sequencing oligos - sequence (5′-3′)** | | |
| Bace1_Ex6_GT-for | ATCTGCCAGGACCTACTTA | |
| Bace1_Ex6_GT-rev | CTGATGGGAAGGATCTACG | |
| **Off-target analysis oligos - Sequence (5′-3′)** | | |
| Bace1_289-OT1-for | TGGGTGACATTTGGGGTTCA | |
| Bace1_289-OT1-rev | GCCACAGTCAAGTAAGGCCA | |
| Bace1_289-OT2-for | CCCACACTTTCCATTCACATCCA | |
| Bace1_289-OT2-rev | ATGCTGTAGTCCCAAGGTCCATA | |
| Bace1_289-OT3-for | TTGTTTGGTAGCACATCAAATGC | |
| Bace1_289-OT3-rev | TCAATTCCTCTTATTGTACACTGC | |
| Bace1_289-OT4-for | TCAAAAGAAACTACCTCCCTGG | |
| Bace1_289-OT4-rev | CTGCAACTGCAGACAAAGC | |
| Bace1_289-OT5-for | TCCTGCCAACAACTCCTACTAC | |
| Bace1_289-OT5-rev | CAAGGGGTAAAACAAGCGAGC | |
| Bace1_289-OT6-for | ATCAGTCGGCTCATCACACC | |
| Bace1_289-OT6-rev | CCTCCATGGTTCTCCACCAC | |
| Bace1_289-OT7-for | AGAAAACCTTGCAGGGAGGG | |
| Bace1_289-OT7-rev | CCATCATCACCTCCTGGCTC | |
| Bace1_289-OT8-for | AGGTTGCCTTAGCTGTTGCT | |
| Bace1_289-OT8-rev | GGATGCGGAGTGGGATACTG | |
| Bace1_289-OT9-for | CTGACTGACGTACGGACAAGTTT | |
| Bace1_289-OT9-rev | TAAGCTTCCCTTCCTCCACCAAC | |
| Bace1_289-OT10-for | AAGCATTCACTCTAGGTAGGGC | |
| Bace1_289-OT10-rev | ATCAGCAATGCGGGTCCAT | |
| **Chemicals, enzymes and other reagents** | | |
| sAPPβ ELISA kit | IBL | JP27416 |
| BCA Protein Assay | Uptima Interchim | UP95425 |
| Papain (200 U) | Sigma-Aldrich | P3125 |
| L-Cysteine | Sigma-Aldrich | C6852 |
| Poly-ᴅ-lysine | Sigma-Aldrich | P6407 |
| DMEM | Thermo Fisher | 61965026 |
| FBS | Sigma-Aldrich | F1283 |
| Pen/Strep | Thermo Fisher | 15070063 |

| Reagent/resource | Reference or source | Identifier or Catalog Number |
|---|---|---|
| GlutaMAX | Thermo Fisher | 35050061 |
| Ac4ManNAz | Thermo Fisher | C33366 |
| Neurobasal medium | Thermo Fisher | 21103049 |
| B27 | Thermo Fisher | 17504044 |
| Protease inhibitor | Sigma-Aldrich | P8340 |
| PBS buffer | Thermo Fisher | 10010015 |
| XE991 | Sigma-Aldrich | X2254 |
| Picrotoxin | Sigma-Aldrich | P1675 |
| Kynurenic acid | Sigma-Aldrich | K3375 |
| Mounting Medium | Carl Roth | HP20.1 |
| Poly-D-lysine-coated slides | VWR | 631-0107 |
| Tissue-Tek | Sakura Finetek | 4583 |
| **Software** | | |
| ImageJ | NIH | Version 1.53t |
| GraphPad Prism | GraphPad Software | Version 8.4.2 |
| DIA-NN | Demichev et al, 2020 | Version 1.8 |
| Perseus | Tyanova et al, 2016 | Version 1.6.2.3 |
| OriginPro | OriginLab | Versions 2018G, 2021 & 2023 |
| UniProt REST API | UniProt Consortium, 2023 | - |
| Phobius web server via REST API | https://www.ebi.ac.uk/Tools/services/rest/phobius | - |
| Python code | This study, inspired by the QARIP web server | Python 3.9 |
| Clampfit | Molecular Devices | 10.6 and 11.2 |
| EGErA sleep recording software | SEA, Cologne, Germany | V1 and V2 |
| Sleep analysis / MATLAB | Kreuzer et al, 2015; Uygun et al, 2019; Altunkaya et al, 2024; Joyce et al, 2024; | R2023a |
| **Other** | | |
| ImageQuant LAS 4000 Mini | GE Healthcare | - |
| nanoElute nanoHPLC | Bruker | - |
| timsTOF Pro + CaptiveSpray source | Bruker | - |
| Vanquish Neo nanoLC | Thermo Fisher Scientific | - |
| Exploris 480 mass spectrometer | Thermo Fisher Scientific | - |
| PepSep C18 column (15 cm × 75 µm ID) | Bruker | - |
| Multiclamp 700B | Molecular Devices | - |
| MiniDigi | Molecular Devices | - |
| CM 3050S cryostat | Leica | |
| AxioObserver.Z1 with LSM780 | Zeiss | |
| Motor test equipment | This study/see videos | Custom made |
| 8-pin PCB-sockets | PRECI-DIP, Delemont, Switzerland | 801-87-010-10-001101 |
| Weight-balanced swivel system | MPI for Psychiatry, Munich, Germany | Custom made |
| EEG recording system | MPI for Psychiatry, Munich, Germany | Custom made |
| Commutators | MPI for Psychiatry, Munich, Germany | Custom made |
| Recording cages | MPI for Psychiatry, Munich, Germany | Custom made |

## BACE1[em2Bwef] (Bace1 D289N knock-in, BACE1-KI) mice

The BACE1[em2Bwef] mutant mouse line was generated by CRISPR/Cas9-assisted gene editing in mouse zygotes as described previously (Wefers et al, 2023). Briefly, pronuclear stage zygotes were obtained by mating C57BL/6J males with superovulated C57BL/6 J females (Charles River). Embryos were then microinjected into the male pronucleus with a *Bace1*-specific CRISPR/Cas9 ribonucleoprotein

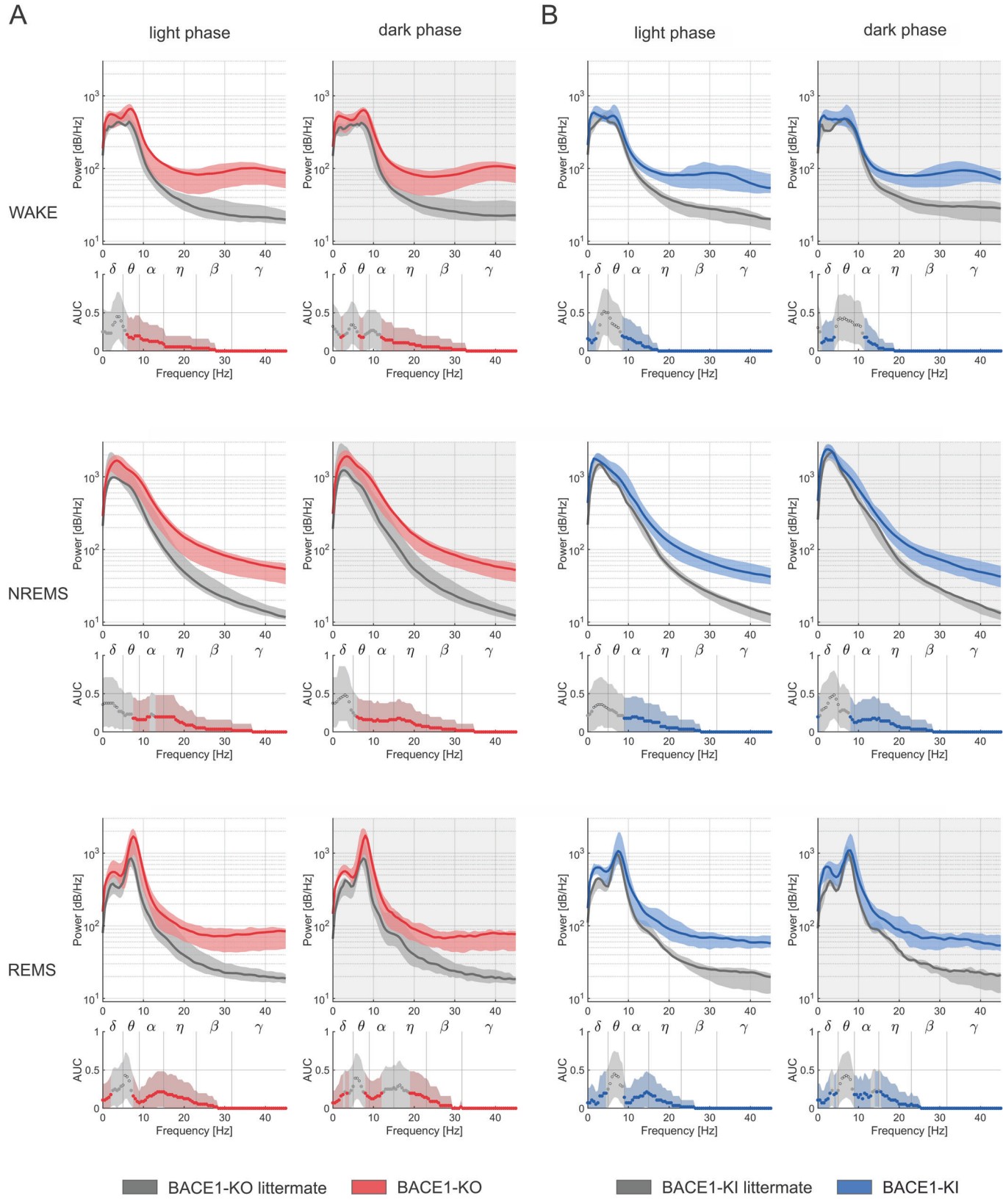

**Figure 5.   The EEG power spectrum was increased in transgenic BACE1-KO and BACE1-KI mice.**

(A, B) EEG recordings were collected from freely moving animals over a 23-hour period, encompassing a 12 h light phase and a 12 h dark phase (11 h recorded). Sleep stages —WAKE, non-REM sleep (NREMS), and REM sleep (REMS)—were identified based on EEG spectral features and concurrent EMG data. Analyses were conducted separately for the dark and light phases. The power spectrum was computed using a fast Fourier transform (FFT). Data were presented as median power spectral density (PSD) in dB/Hz (log-scale), with shaded bands indicating the bootstrapped 95% confidence intervals (CI) derived from 10,000 resamples. Sample sizes: BACE1-KO ($n = 8$) vs. WT littermates ($n = 7$), and BACE1-KI ($n = 8$) vs. WT littermates ($n = 7$).

(RNP) solution consisting of 50 ng/μL SpCas9 protein (IDT), 0.6 μM crRNA (IDT), 0.6 μM tracrRNA (IDT), and 25 ng/μL mutagenic single-stranded oligodeoxynucleotide (ssODN) ssBace1-D289N, (5' to 3'):

GATTCACCCCTCCCACCTCTCCCCCCCTCTCTCCTCAGT ACAACTACGACAAGAGCATTGTcaACAGTGGGACCACCAA CCTTCGCTTGCCCAAGAAAGTATTTGAAGCTGCCGTCAAG TCC

comprising the p.D289N (GAC > AAC) substitution and an additional silent mutation for genotyping purposes (IDT; Fig. 1A). After microinjection, treated zygotes were transferred into pseudo-pregnant CD-1 foster animals. Mutant founder animals were crossed to C57BL/6 J animals to isolate the respective allele, and validation of the targeted locus was performed on genomic DNA from F1 animals by RFLP and Sanger sequencing. To exclude additional unwanted modifications, putative off-target sites of the *Bace1*-specific crRNA were predicted using the CRISPOR online tool (Concordet and Haeussler, 2018). Predicted loci (Top 10) were PCR-amplified and verified by Sanger sequencing. Validated animals without any off-target mutations were used for further breeding.

All animal experiments were approved by the Bavarian government under license number ROB-55.2Vet-2532.Vet_02-16-121. All mice were handled according to institutional guidelines approved by the Animal Welfare and Use Committee of the Government of Upper Bavaria and housed in standard cages in a specific pathogen-free facility on a 12 h light/dark cycle with ad libitum access to food and water.

### Genotyping of BACE1$^{em2Bwef}$ mice

For genotyping of BACE1$^{em2Bwef}$, the tails of mice were cut and lysed with lysis buffer (containing 25 mM NaOH and 0.2 mM ethylene-diaminetetraacetic acid [EDTA], pH 12). Then the genome was precipitated by isopropanol and washed with 70% ethanol. One microliter of the DNA template was used along with both the wild-type and mutant primer sets for PCR. Wild-type set of primers (5'-ATCTGCCAGGACCTACTTAGTGCTT -3' and 5'- GGTTGGT GGTCCCACTGTCC -3') and mutant set of primers (5'- ATCTGC CAGGACCTACTTAGTGCTT -3' and 5'- GGTTGGTGGTCCCA CTGTTG -3') yield a 260 bp amplicon.

### BACE1$^{tm1Psa}$ (Bace1 knockout, BACE1-KO) mice

BACE1$^{tm1Psa}$ mice were generated by inserting a neomycin cassette into exon 1 of the Bace1 gene, resulting in a premature stop codon (Dominguez et al, 2005). These mice were backcrossed onto a C57BL/6 J background for over 15 generations. PCR amplification was employed to detect the wild-type allele or the neomycin cassette. All aspects of housing, feeding, breeding, and handling were conducted in accordance with federal and institutional

guidelines, with approval from the local government of Unter-franken. Experimental procedures included mice of both sexes.

### BACE1$^{tm1Pcw}$ (Bace1 knockout, BACE1-KO) mice

BACE1-KO mice were obtained from Jackson Laboratory (Bar Harbor, ME, USA, strain B6.129-Bace1$^{tm1Pcw/J}$) (Cai et al, 2001). Tissue from Bace1$^{tm1Pcw/J}$ mice was used exclusively as a comparator in the immunoblot analyses.

### Preparation of brain fractions

Mouse brains were isolated from 3-month-old BACE1$^{em2Bwef}$ (wild-type (WT), heterozygous (HET) and BACE1-KI), and BACE1$^{tm1Psa}$ (WT and BACE1-KO) mice. Soluble (DEA) and membrane proteins were processed as previously described (Kuhn et al, 2012). Protein concentrations in the fractions were quantified with the BCA assay (Uptima Interchim, UP95425), and 20 μg protein was used for further Western Blot analysis.

### Immunoblot analysis

Antibodies against BACE1 (D10E5, Cell Signaling Technology, #5606), APP (22C11, Merck Millipore, MAB348), soluble SEZ6 (polyclonal antibody, kindly provided by Dr. Jenny Gunnersen), NrCAM (Abcam, ab24344), β-actin (Sigma-Aldrich, A5316), and Calnexin (Enzo Life Sciences, ADI-SPA-860) were used. HRP-conjugated anti-mouse and anti-rabbit secondary antibodies (Dako) were utilized for detection.

For analysis of soluble and membrane fractions, 20 μg of total protein were separated by SDS–PAGE using 10% tris-glycine gels and transferred onto PVDF membranes. Membranes were blocked with 3% BSA in PBS containing 0.01% Tween-20, incubated overnight at 4 °C with primary antibodies, washed, and then incubated with the appropriate HRP-conjugated secondary antibody. Signal detection was performed using the ImageQuant LAS 4000 Mini system (GE Healthcare), and band intensities were quantified using ImageJ software (version 1.53t). Statistical analysis was conducted with GraphPad Prism.

### sAPPβ measurement

sAPPβ in the soluble fraction of mouse brain was quantified using a sAPPβ ELISA kit (JP27416, IBL) according to the manufacturer's protocol.

### Primary neuron culture

Primary cortical neurons were isolated from embryonic day 16 (E16) embryos of BACE1$^{em2Bwef}$ HET female mice mated with HET male mice as described before (Colombo et al, 2013). Briefly, embryonic

cortices were collected and subjected to incubation in a digestion medium composed of DMEM containing 200 U of papain and 1 mg/mL L-cysteine, with a pH of 7.4. The incubation was carried out at 37 °C for 20 min. Subsequently, the digested tissues underwent mechanical dissociation in a plating medium consisting of DMEM containing 10% FBS and 1% penicillin/streptomycin. The dissociated cells were then plated onto 6-well plates coated with 25 µg/mL of poly-D-lysine. The plating density was set at $1.6 \times 10^6$ cells per well. Four hours after plating, the medium was replaced with Neurobasal medium supplemented with 2% B27, 1% penicillin/streptomycin, and 0.5 mM GlutaMAX. The neurons were maintained in a cell culture incubator with a temperature of 37 °C and a 5% $CO_2$ concentration.

## Mass spectrometry (MS) sample preparation (high-performance secretome protein enrichment with click sugars (hiSPECS) method and lysate)

For hiSPECS samples, primary cortical neurons at DIV4 were cultured for another 48 h in neuronal medium supplemented with 50 µM Ac$_4$ManNAz or DMSO as a control. At DIV6, the supernatant was filtered and stored with protease inhibitors for further secretome protein enrichment and digestion as described previously (Tüshaus et al, 2020).

For neuronal lysate samples, after the supernatant was removed, the remaining neurons were washed three times with PBS, lysed with STET buffer, and cleared by centrifugation for 15 min at 16,000×g at 4 °C, followed by further tryptic digestion and peptide purification using the single-pot solid-phase sample preparation 3 protocol as previously described (Hughes et al, 2019).

## Liquid chromatography tandem mass spectrometry

The hiSPECS peptides were dissolved in 20 µL 0.1% formic acid. A volume of 8 µL was separated on a nanoElute nanoHPLC system (Bruker) on an in-house packed C18 analytical column (15 cm × 75 µm ID, ReproSil-Pur 120 C18-AQ, 1.9 µm, Dr. Maisch GmbH) with a gradient of water and acetonitrile (B) containing 0.1% formic acid at a flow rate of 300 nL/min (0 min, 2% B; 2 min, 5% B; 62 min, 24% B; 72 min, 35% B; 75 min, 60% B) and a column temperature of 50 °C. The nanoHPLC was online coupled to a timsTOF pro mass spectrometer (Bruker) with a CaptiveSpray ion source (Bruker). A data-independent acquisition parallel accumulation–serial fragmentation (diaPASEF) method was used for spectrum acquisition. The ramp time was set to 100 ms. The diaPASEF method included one full scan and 13 scans with two m/z windows per diaPASEF scan, with a width of 27 m/z, resulting in a cycle time of 1.4 s.

The lysate peptides were dissolved in 20 µL 0.1% formic acid. A volume of 3 µL was separated on a Vanquish Neo nanoLC system coupled online to an Exploris 480 mass spectrometer (Thermo Fisher Scientific, Waltham, MA). Peptides were separated using a PepSep C18 column (15 cm × 75 µm ID, Bruker) with a 72 min binary gradient of water and acetonitrile (B) containing 0.1% formic acid at a flow rate of 300 nL/min (0 min, 6% B; 2 min, 6% B; 62 min, 31% B; 67 min, 50% B; 72 min, 95% B). Data-independent acquisition (DIA) was performed using an MS1 full scan followed by 20 variable DIA windows. Full scans were acquired with 120,000 resolution, automatic gain control (AGC) of 300% from the normalized AGC target, and maximum injection time of 100 ms.

Subsequently, 20 isolation windows were scanned with a resolution of 30,000, an AGC of 3000% and the maximum injection time was set to auto to achieve the optimal cycle time. Collision-induced dissociation fragmentation was performed with 25, 27.5, and 30% of the normalized HCD collision energy. Data acquisition was carried out with a FAIMS device in standard resolution mode, using a total carrier gas flow of 3.5 L/min and compensation voltage of −45 V.

## Proteomic data analysis

The software DIA-NN version 1.8 was used to analyze the data and to perform protein label-free quantification (LFQ) (Demichev et al, 2020). The raw data were searched against a single-protein-per-gene database from Mus musculus (UniProt, 21,994 entries, download: 2022-01-25) combined with a database of common protein contamination from MaxQuant (246 entries) using a library free search. Trypsin was used as a protease with a maximum of two missed cleavages. Acetylation of protein N-termini and oxidation of methionines were set as variable modifications. Carbamidomethylation of cysteines was defined as a fixed modification. The precursor and fragment ion m/z ranges were limited from 350 to 1002 and 200 to 1700, respectively. An FDR (false discovery rate) threshold of 1% was applied for peptide and protein identifications. The mass accuracy was set to 15 ppm for both MS1 and MS2 scans. The ion mobility windows were automatically adjusted by the software. The match between runs option was enabled.

The statistical analysis was performed with the software Perseus version 1.6.2.3 (Tyanova et al, 2016). The protein LFQ intensities were log2-transformed. A two-sided Student's t-test was applied to detect significant protein abundance changes. Three valid quantification values were required per group for statistical testing. Additionally, a permutation-based false discovery rate estimation was applied with an FDR of 5% at s0 = 0.1 as the threshold (Tusher et al, 2001).

## Peptide localization mapping (Supplementary method)

To determine whether the identified peptides from the selected 15 BACE1 substrates originated from extracellular domains, we aligned peptide sequences with the full-length protein sequences and visualized their localization on domain-annotated graphical representations. Only unique peptides were included in the analysis. Protein sequences and domain information were obtained from UniProt using programmatic access to the UniProt REST API (https://rest.uniprot.org/uniprotkb/search) (UniProt Consortium, 2023), with specific return fields including sequence, length, ft_signal, ft_topo_dom, ft_transmem, ft_intramem, and ft_propep (see also https://www.uniprot.org/help/return_fields).

In cases where UniProt domain annotations did not cover the entire protein sequence, additional domain information was inferred using the Phobius web server (https://phobius.sbc.su.se/) via its REST API (https://www.ebi.ac.uk/Tools/services/rest/phobius). If a protein's GI number could not be matched to a UniProt accession number, we selected the longest available isoform; if peptides mapped to multiple isoforms, the analysis was conducted across all available isoforms. For proteins not found in UniProt, or when peptide sequences did not align fully with the UniProt reference, we utilized the RefSeq database in combination with Phobius predictions.

Peptide localization mapping was performed in Python 3.9 with the pandas (1.1.5), numpy (1.25.2), requests (2.28.1), seaborn (0.12.2), and matplotlib (3.7.2) libraries. This analytic approach was inspired by the QARIP web server (https://webclu.bio.wzw.tum.de/qarip/) (Ivankov et al, 2013).

## Electrophysiological slice recordings

Brain slices were prepared from mice aged 3–5 months under isoflurane anesthesia, as described previously (Hartmann et al, 2018; Hessler et al, 2015). Briefly, transverse dorsal hippocampal slices (350 μm) were cut in ice-cold sucrose-based artificial cerebrospinal fluid (aCSF), containing (in mM) 75 sucrose, 87 NaCl, 2.5 KCl, 0.5 $CaCl_2$, 7 $MgCl_2$, 1.25 $NaH_2PO_4$, 25 $NaHCO_3$, and 10 D-glucose, and then incubated in the same solution for 10 min at 35 °C. Slices were kept at room temperature for at least 2 h in aCSF containing (in mM) 125 NaCl, 3 KCl, 1 $CaCl_2$, 3 $MgCl_2$, 1.25 $NaH_2PO_4$, 25 $NaHCO_3$, and 10 D-glucose. Individual slices were then transferred to a submerged recording chamber (perfused with aCSF containing 2.5 mM $CaCl_2$ and 1.5 mM $MgCl_2$ at $31 \pm 1$ °C) that was mounted on the stage of an upright microscope. All solutions were gassed with 95% $O_2$/5% $CO_2$.

Whole-cell current-clamp recordings were performed in CA1 pyramidal cells and dentate gyrus (DG) granule cells (GC) located in the outer part of the granule cell layer (i.e., close to the molecular layer, to avoid sampling immature GC). Patch pipettes were filled with (in mM) 135 K-gluconate, 5 HEPES, 3 $MgCl_2$, 5 EGTA, 2 $Na_2ATP$, 0.3 $Na_3GTP$, 4 NaCl (pH 7.3). After obtaining a stable reading of the cells' resting membrane potential (RMP), the membrane potential was always adjusted to $-70$ mV by DC current injection for the sake of comparability before the M-current blocker XE991 (10 μM) was applied. The magnitude of the XE991-induced depolarization was given by the shift of the membrane potential from baseline ($-70$ mV) to the drug's maximal effect recorded 4–8 min after onset of application. In some experiments, DG granule cells were pharmacologically isolated from GABAergic and glutamatergic synaptic inputs by picrotoxin (100 μM) and kynurenic acid (2 mM), respectively. All potentials were corrected for liquid junction potential. Signals were filtered at 6 kHz and sampled at 20 kHz using a Multiclamp 700B amplifier in conjunction with Digidata 1550 interface and pClamp10.6 software (Molecular Devices). MiniDigi 1 A and AxoScope 10.6 were used for low-resolution scope recording, sampled at 1 kHz. Data analysis was performed offline with Clampfit 10.6. Drugs and chemicals were obtained from Tocris Bioscience and Sigma-Aldrich. Data were expressed as mean ± SEM. Statistical comparisons of data were performed with OriginPro 2018G (OriginLab), using ANOVA or Student's t-test as appropriate.

## Immunohistochemistry and confocal imaging

Mice at postnatal day 40 (P40) were anesthetized with isoflurane (Piramal) and euthanized by decapitation. Whole brains, cochleae from both temporal bones, and sciatic nerves were dissected. Cochleae were placed in 4% paraformaldehyde (PFA) in phosphate-buffered saline (PBS) at 4 °C for 2 h following the creation of a small apical hole. For whole-mount immunostaining, the apical portion of the organ of Corti was dissected from the modiolus, stria vascularis, and tectorial membrane. Samples were blocked and permeabilized at room temperature for 1 h in a blocking solution (NGSB) consisting of 10% normal goat serum, 0.3% Triton X-100, 20 mM phosphate buffer, and 450 mM NaCl. Primary antibodies, rabbit anti-NF200 (Sigma-Aldrich, N4142; 1:600) and mouse anti-MBP (Santa Cruz Biotechnology, sc-271524; 1:400), were applied overnight at 4 °C in blocking solution. Secondary antibodies, goat anti-rabbit Cy3 (Jackson ImmunoResearch, 111-165-144; 1:200) and goat anti-mouse Alexa Fluor 488 (Molecular Probes, A-11029; 1:500), were applied at room temperature for 90 min. Specimens were mounted using DAPI-containing medium (Carl Roth) on poly-D-lysine-coated slides (VWR) and stored at 4 °C until imaging.

Dissected brains and sciatic nerves were fixed in 4% PFA in PBS at 4 °C for 3 and 24 h, respectively. Tissues were cryopreserved in 20% (w/v) sucrose for 24 h, embedded in Tissue-Tek (Sakura Finetek), and snap-frozen in methylbutane at $-40$ °C for 90 s. Coronal brain sections (14 μm) and cross-sections of sciatic nerves (14 μm) were prepared using a Leica CM 3050S cryostat, mounted on poly-D-lysine-coated slides (VWR), and stored at $-20$ °C. Brain sections were rehydrated with PBS and permeabilized with PBS containing 0.5% Triton X-100 for 30 min. Blocking was performed for 90 min in PBS with 1% bovine serum albumin (BSA, w/v), 5% normal donkey serum, and 0.1% Triton X-100. Sections were incubated with rabbit anti-synaptoporin (SynapticSystems, 102 002; 1:500) or rabbit anti-BACE (Cell Signaling, #5606; 1:250) antibodies at room temperature for 24 h. Secondary antibody incubation (donkey anti-rabbit Alexa Fluor 488, Molecular Probes, A-21206; 1:500) was performed at 4 °C for 24 h. Sciatic nerve sections were processed similarly to the cochlear whole- mounts, using the same blocking and antibody protocols. After immunostaining, slides were rinsed with PBS, mounted with DAPI-containing medium (Carl Roth), sealed, and stored at 4 °C.

Images were acquired using an AxioObserver.Z1 equipped with an LSM780 confocal module (Zeiss), 405 nm laser diode, argon laser LGN 3001 (LASOS), DPSS 561-10, Plan-Apochromat 63×/1.40 oil DIC (Zeiss), and EC Plan-Neofluar 10×/0.3 (Zeiss) objectives, managed by ZEN 2010 software (Zeiss).

To measure the infrapyramidal bundle (IPB) length, brains from BACE1-WT and BACE1-KI mice were analyzed. Six coronal sections from each brain were selected using anatomical landmarks such as hippocampal and ventricular morphology. IPB length was defined as the distance through the center of synaptoporin-positive axons beneath the pyramidal cell layer, measured from the end of the pyramidal cell layer to the most distal point of uninterrupted synaptoporin labeling. The superior pyramidal bundle (SPB) and stratum lucidum (SL) length were measured as the distance through the center of synaptoporin-positive axons above the pyramidal cell layer and through the SL, extending to the distal end. IPB lengths were normalized to the total SPB + SL length for each section, and the ratios were averaged over six sections per brain.

## Motor coordination tests and analysis

Motor coordination was assessed in mice aged 4–6 months following a three-day handling period. The study cohort included BACE1-KO and littermate WT, and BACE1-KI and littermate WT mice of mixed-sex. BACE1-KO: WT (four males/four females), KO (four males/four females); BACE1-KI: WT (four males/three females), KI (five males, three females). We did not observe any

significant sex-related differences. The setup was similar to a previously described method (Chen et al, 2010). Footprint pattern: Mice walked across a clear plate while paw positions were recorded using a video camera placed underneath the plate. Footprint patterns were analyzed to determine the positions of the hindpaws, specifically whether more than 50% of the paw area was located outside the mouse's body line. For each mouse, ten hindpaw positions were analyzed. Ladder Walking Test: Mice were tested on a horizontal ladder consisting of irregularly spaced rods with a diameter of 2 mm (max distance 20 mm). Each animal walked across the ladder until at least 60 steps were recorded. Video recordings of the runs were analyzed to count slips of the hindpaws (Data have been deposited, see Data availability). Balance Beam Test: Mice were tested on two $50 \times 1$ cm beams (circular and rectangular cross-section) elevated above the ground between two platforms. Animals were required to walk across the beam until a minimum of 20 walking steps had been documented. Video recordings were analyzed to quantify slips of the hindpaws (Data have been deposited, see Data availability). All video analyses were performed by an investigator blinded to the experimental conditions. All behavioral experiments were approved by the local government of Unterfranken.

## Electrode assembly, implantation surgery, and EEG recordings

Electrode sockets were custom-built by soldering 751 GG gold wire (ø 150 μm) onto 8-pin PCB-sockets (PRECI-DIP, Delemont, Switzerland). Each socket held two EEG, two EMG, and one ground electrode, with a dental cement cover providing additional rigidity and insulation.

Mice were first anesthetized with 5% isoflurane in a plexiglass chamber and then transferred to a stereotact where anesthesia was maintained at 1.5–2.2% (flow rate ~190 ml/min). For preoperative analgesia (24 h), the animals received carprofen (0.067 mg/kg) in the drinking water. Body temperature was regulated at 37 °C on a feedback-controlled heating pad. A subcutaneous injection of lidocaine (≤10 mg/kg) was then administered, and the surgical site was shaved. The scalp was then incised, and the periosteum removed. Skull penetrations (Ø500 μm) were drilled for electrode placement and jeweler's screws, with EEG electrodes positioned on the dura and EMG electrodes inserted bilaterally into the musculus semispinalis capitis. The implants were secured with dental cement, and the wound was closed with single-button sutures. Postoperative care included monitoring until recovery (minimum 15 min), a heated cage, and carprofen (0.067 mg/kg) in the drinking water 4 days post-surgery. Recordings were started 10 days after surgery.

The implanted electrode socket was connected via a custom cable to a commutator and a weight-balanced swivel system (MPI for Psychiatry, Munich, Germany). EEG signals were amplified by 10,000× and EMG by 1,000× (analog bandpass filter: 0.5–120 Hz, Prescisor Messtechnik, Munich, Germany) and digitized at a minimum of 256 Hz (MPI for Psychiatry, Munich, Germany). Chronic EEG recordings were conducted for 23 h per day over three consecutive days, consisting of 12 h of light (Lights ON) and 11 h of darkness (Lights OFF). The final hour of each dark phase was reserved for animal care (under red light), maintenance of the setup, and offline processing of the daily EEG data using the EGErA sleep scoring software (Cologne, Germany) (Altunkaya et al, 2024;

Fenzl et al, 2007, 2011; Kreuzer et al, 2015; Polta et al, 2013; Romanowski et al, 2010; Touma et al, 2009). Recordings were carried out in batches consisting of at least two mice at a time. Sex distribution for each group is detailed above. Although some tests related to sex differences reached statistical significance, their number was consistent with the false-positive rate expected by chance at the $\alpha = 0.05$ level. For further details, please refer to (Altunkaya et al, 2024). All behavioral experiments were approved by the government of Bavaria.

## Data analysis: sleep–wake behavior

Raw EEG/EMG data (.tdms files) were downsampled to 256 Hz, then divided into light (ZT0-12) and dark (ZT12-23) periods. Frequency bands (δ: 0.5–5 Hz; θ: 6–9 Hz; α: 10–15 Hz; η: 16–22.75 Hz; β: 23–31.75 Hz; γ: 31.75–45 Hz) were extracted (Fenzl et al, 2007), and sleep–wake states were manually scored in 4 s epochs using a semi-automated scoring software (Kreuzer et al, 2015). A state change was defined if the subject sustained a given vigilance state for at least three consecutive epochs. Minimal movement artifacts (<0.015% of data) were manually thresholded and excluded.

Vigilance state profiles were generated by computing group median values in 2 h bins, with 10,000-fold bootstrapped confidence intervals. Bout durations were expressed as the median duration per vigilance state for light and dark phases. State transition probabilities were calculated for naturally occurring transitions (e.g., WAKE ▸ NREMS, NREMS ▸ REMS, etc.).

Normalized power spectral density (PSD) plots were computed from the EEG data. Bootstrapping (10,000 iterations) was used to estimate variability and generate 95% confidence intervals, with results expressed as percentages.

Sleep spindles during NREMS were automatically detected using a MATLAB algorithm (Uygun et al, 2019). EEG traces were bandpass filtered (10–15 Hz) with a Butterworth filter, and the root mean square (RMS) was calculated using a 750 ms window. RMS values were cubed to enhance the signal-to-noise ratio, and spindles were identified using a two-threshold approach (lower: 1.0, upper: 2.5 times the mean cubed RMS). Spindle parameters, including duration (0.5–2 s), normalized amplitude, amount, and density (spindles per minute), were then quantified. Spectral differences were evaluated by calculating the area under the ROC curve (AUC) for each frequency bin, with 10,000-fold bootstrapped 95% confidence intervals. For all significance tests used, alpha was set to 0.05. For further details pertaining to all implantation and data analysis methods, please refer to (Altunkaya et al, 2024; Joyce et al, 2024).

## Experimental design and statistical analysis

Data analysis and statistics were performed using OriginPro (version 2021 and 2023, OriginLab). Numbers are given as mean ± SEM, if not stated otherwise. Normal distribution was tested using the Kolmogorov–Smirnov test. The number of replicates and tests to determine statistical significance are stated in the text and figure legends of the respective experiments.

## Use of AI tools

OpenAI's ChatGPT (GPT-4; accessed during March–April 2025) was used solely for language editing—to improve clarity, grammar, and readability. All substantive content, analyses, and

interpretations were independently conceived and verified by the authors, who accept full responsibility for the manuscript.

## Data availability

The mass spectrometry proteomics data have been deposited to the ProteomeXchange Consortium via the PRIDE (Perez-Riverol et al, 2025) partner repository with the dataset identifier PXD065011. The remaining data have been deposited in Zenodo (https://doi.org/10.5281/zenodo.16910972).

The source data of this paper are collected in the following database record: biostudies:S-SCDT-10_1038-S44319-025-00604-4.

## Peer review information

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

## Acknowledgements

We are grateful to Iwona Izydorczyk, Birgit Vogler, and Anna Berghofer for technical assistance, and to Merav Shmueli and Jasenka Njavro for help with the initial characterization of the BACE1-KI mouse line. This study was supported by the Deutsche Forschungsgemeinschaft (DFG, German Research Foundation, HU 2358/1-1) to TH and PO1799/6-1 to AP. This work was also funded by the Deutsche Forschungsgemeinschaft (DFG, German Research Foundation) under Germany's Excellence Strategy within the framework of the Munich Cluster for Systems Neurology (EXC 2145 SyNergy—ID 390857198) and by the BMBF (FKZ161L0214C, ClinspectM) to SFL. We gratefully acknowledge support to H.H. from the Interdisziplinäres Zentrum für Klinische Forschung (IZKF), Universitätsklinikum Erlangen, Germany. The present work was performed in (partial) fulfillment of the requirements for obtaining the degree Dr. med.

## Author contributions

**Hannah Heininger**: Formal analysis; Investigation; Visualization. **Xiao Feng**: Formal analysis; Investigation; Visualization. **Alp Altunkaya**: Data curation; Software; Formal analysis; Visualization. **Fang Zheng**: Formal analysis; Investigation; Visualization. **Florian Stockinger**: Formal analysis; Investigation; Visualization. **Benedikt Wefers**: Resources; Investigation. **Stephan A Müeller**: Data curation; Formal analysis; Methodology. **Pieter Giesbertz**: Formal analysis; Investigation. **Sarah K Tschirner**: Investigation. **Dorina Shqau**: Investigation. **Helmuth Adelsberger**: Resources; Formal analysis; Methodology. **Alexey Ponomarenko**: Resources; Methodology; Writing—review and editing. **Thomas Fenzl**: Conceptualization; Resources; Supervision; Methodology. **Christian Alzheimer**: Conceptualization; Funding acquisition; Writing—original draft. **Stefan F Lichtenthaler**: Conceptualization; Supervision; Funding acquisition; Visualization; Methodology; Writing—original draft. **Tobias Huth**: Conceptualization; Formal analysis; Supervision; Funding acquisition; Visualization; Writing—original draft; Project administration.

Source data underlying figure panels in this paper may have individual authorship assigned. Where available, figure panel/source data authorship is listed in the following database record: biostudies:S-SCDT-10_1038-S44319-025-00604-4.

## Funding

## Disclosure and competing interests statement

The authors declare no competing interests.

