## [Peer Review File · EMBO Reports]

BACE1 regulates sleep-wake cycle through both enzymatic and non-enzymatic actions.

Hannah Heininger, Xiao Feng, Alp Altunkaya, Fang Zheng, Florian Stockinger, Benedikt Wefers, Stephan Mueller, Pieter Giesbertz, Sarah Tschirner, Dorina Shqau, Helmuth Adelsberger, Alexey Ponomarenko, Thomas Fenzl, Christian Alzheimer, Stefan Lichtenthaler, and Tobias Huth

Corresponding author(s): Tobias Huth (tobias.huth@fau.de) , Stefan Lichtenthaler (stefan.lichtenthaler@dzne.de)

Review Timeline:

Submission Date:	25th Apr 25
Editorial Decision:	10th Jun 25
Revision Received:	21st Aug 25
Editorial Decision:	12th Sep 25
Revision Received:	26th Sep 25
Accepted:	7th Oct 25

Editor: Esther Schnapp

Transaction Report:

Dear Dr. Huth,

Thank you for your patience while your manuscript was peer-reviewed at EMBO reports. We have now received the full set of referee reports that is pasted below.

As you will see, the referees acknowledge that the findings are interesting. However, they also have some suggestions for how the study should be strengthened. I think all suggestions are good and should be addressed. Please let me know in case you disagree and we can discuss the exact revision requirements further, also in a video chat, if you like.

I would thus like to invite you to revise your manuscript with the understanding that the referee concerns must be fully addressed and their suggestions taken on board. Please address all referee concerns in a complete point-by-point response. Acceptance of the manuscript will depend on a positive outcome of a second round of review. It is EMBO reports policy to allow a single round of major revision only and acceptance or rejection of the manuscript will therefore depend on the completeness of your responses included in the next, final version of the manuscript.

We realize that it is difficult to revise to a specific deadline. In the interest of protecting the conceptual advance provided by the work, we recommend a revision within 3 months (10th Sep 2025). Please discuss the revision progress ahead of this time with the editor if you require more time to complete the revisions.

You can either publish the study as a short report or as a full article. For short reports, the revised manuscript should not exceed 29,000 characters (including spaces but excluding materials & methods and references) and 5 main plus 5 expanded view figures. The results and discussion sections must further be combined, which will help to shorten the manuscript text by eliminating some redundancy that is inevitable when discussing the same experiments twice. For a normal article there are no length limitations, but it should have more than 5 main figures and the results and discussion sections must be separate. In both cases, the entire materials and methods must be included in the main manuscript file.

- 1) A data availability section providing access to data deposited in public databases is missing. If you have not deposited any data, please add a sentence to the data availability section that explains that.
- 2) Your manuscript contains statistics and error bars based on $n=2$. Please use scatter blots in these cases. No statistics should be calculated if $n=2$.

5) a complete author checklist, which you can download from our author guidelines . Please insert information in the checklist that is also reflected in the manuscript. The completed author checklist will also be part of the RPF.

6) Please note that all corresponding authors are required to supply an ORCID ID for their name upon submission of a revised

manuscript (). Please find instructions on how to link your ORCID ID to your account in our manuscript tracking system in our Author guidelines

- the name of the statistical test used to generate error bars and P values,
- the number (n) of independent experiments (please specify technical or biological replicates) underlying each data point,
- the nature of the bars and error bars (s.d., s.e.m.),
- If the data are obtained from n {less than or equal to} 2, use scatter blots showing the individual data points.

12) All Materials and Methods need to be described in the main text using our 'Structured Methods' format, which is required for all research articles. According to this format, the Methods section includes a separate Reagents and Tools Table file (listing key reagents, experimental models, software and relevant equipment and including their sources and relevant identifiers) and a Methods and Protocols section describing the methods using a step-by-step protocol format. The aim is to facilitate adoption of the methodologies across labs. More information on how to adhere to this format as well as a downloadable template (.docx) for the Reagents and Tools Table can be found in our author guidelines: <https://www.embopress.org/page/journal/14693178/authorguide#structuredmethods>.

An example of a Method paper with Structured Methods can be found here: <https://www.embopress.org/doi/full/10.1038/s44320-024-00037-6#sec-4>

I look forward to seeing a revised form of your manuscript when it is ready.

Referee #1:

In this manuscript, Heininger et al. report that the loss of function of BACE1 results in alterations in the sleep-wake cycle. The authors generated a proteolysis-deficient BACE1 knock-in mouse and compared it with BACE1 ko mice. Phenotypic analysis of the BACE1 Ki mice revealed similarity with BACE1 KO. BACE1-KI and BACE1-KO mice displayed common and distinct sleep-wake disturbances. Overall, this study was well executed. However, the authors need to address the following points:

- 1) Two different BACE1-KO mouse lines are described in the methods. The authors should clarify why two different lines were used and indicate the specific line used in each experiment for comparison with BACE1-KI mice.
- 2) The sex of the mice was not reported. For the behavioral experiments, sex differences need to be assessed.
- 3) The authors should discuss that most of the phenotypes are common to BACE1-KO and BACE1-KI. The role of non-enzymatic BACE1 is minimal.
- 4) The authors should discuss that a limitation of the study is the use of germline KO and KI mice. Thus, these findings do not provide helpful information for eventual clinical trials with BACE inhibitors

Referee #2:

Hannah Heininger et al.'s study examined the effect of BACE1 on the sleep-wake cycle using BACE1 knockout (BACE1-KO) mice and a newly generated transgenic line that expresses a proteolysis-deficient variant of BACE1 (BACE1-KI). The researchers found that both BACE1-KO and BACE1-KI mice exhibited disturbances in their sleep-wake cycles. They proposed that BACE1 influences sleep architecture and regulates sleep-wake transitions through both proteolytic and non-proteolytic mechanisms. Consequently, the authors suggest that clinical trials involving BACE1-targeted inhibitors should take into account the impact of BACE1 on sleep-modulated behaviors. The findings are significant and relevant for a comprehensive evaluation of the balance between the therapeutic benefits and side effects of BACE1 inhibitors in recent clinical trials. Additionally, the study may provide valuable insights for broader interventions in sleep disorders.

However, certain concerns need to be addressed:

1. The authors should present a quantitative and statistical analysis of protein levels as depicted in Figure 1C.
2. In Figure 2E, the authors should also provide representative traces for BACE1-KO and BACE1-KI. Specifically, is there a difference in resting membrane potential among wild-type (WT), BACE1-KO, and BACE1-KI mice? The authors are also expected to describe their methodology for determining the results of XE991-induced membrane depolarization, as shown in Figure 1F, and explain how they concluded that M-currents were not altered. The reviewer recommends including a detailed description of these issues in the Materials and Methods section.

Referee #3:

The manuscript by Heininger et al reported the characterization of a knock-in mouse model with the mutation of the catalytic site in BACE1 (D289N). Authors in this study showed that this knock-in mice were unable to proteolytic process of BACE1 substrates, as evidenced in Figure 1. These results in Figure 1 by comparing that of BACE1 KO mice confirmed inactivation of proteolytic functions in BACE1 in this KI model. Authors further confirmed hypomyelination phenotypes which were reported to be seen in BACE1 KO mice. Importantly, authors identified clear differences of the hippocampal neuronal KCNQ-mediated M-current of the two mouse lines, in line with their previous finding that the non-enzymatic interaction between BACE1 and KCNQ in the control of M current. Authors in this study attempted to partially attribute this M-current to the observed differential sleep disturbance see in KI vs KO mice. Nevertheless, this study reinforces the proteolytic functions in the regulation of BACE1 in the

modulation of the thalamocortical networks such as sleep spindles in NREMS states. The manuscript is properly written and has a balanced discussion that supports the observations through comparisons of BACE1 KI and KO mice. While I am supportive to publication of this study, the minor concerns may need to be address.

1. Authors only did 24 hrs EEG recording and scored EEG in 12 hrs light phase and 11 hrs dark phase. Usually, it is expected to record mouse EEG for more than 24hrs, and score EEG in 12 hrs light phase and 12 hrs dark phase exclude the first and last 2hrs of the recording data to avoid the artificial interference of mice sleep-wake behavior. It is important to explain the logic of the experimental designs here.

2. In figure 3 compressed EEG traces, NREM time (black spikes) in *Bace1*-ki control is much less than NREM time in BACE1 KI control during 11 hrs dark phase. Also, NREM time is visibly reduced in BACE1-KI compared to BACE1 KO during the dark phase. However, statistical analysis shows no difference between KI and KO mice (Figure 4A vs 4D). This discrepancy should be aware.

We thank the reviewers and the editor for their positive evaluation and their insightful comments.

For the revised version we have now included access to the source data.

- Proteomics (Pride database)

Reviewer access details. Log in to the PRIDE website using the following details:

Project accession: PXD065011

Token: xWkLF18puXgY

Alternatively, reviewer can access the dataset by logging in to the PRIDE website using the following account details:

Username: reviewer_pxd065011@ebi.ac.uk

Password: Ex8PulTeRw1T

- Other data (<https://doi.org/10.5281/zenodo.16910972>)

Response to the Reviewers' Comments

Referee #1:

In this manuscript, Heininger et al. report that the loss of function of BACE1 results in alterations in the sleep-wake cycle. The authors generated a proteolysis-deficient BACE1 knock-in mouse and compared it with BACE1 ko mice. Phenotypic analysis of the BACE1 Ki mice revealed similarity with BACE1 KO. BACE1-KI and BACE1-KO mice displayed common and distinct sleep-wake disturbances. Overall, this study was well executed. However, the authors need to address the following points:

1) Two different BACE1-KO mouse lines are described in the methods. The authors should clarify why two different lines were used and indicate the specific line used in each experiment for comparison with BACE1-KI mice.

The BACE1 KO line BACE1^{tm1Pcw} was used for a single experiment only: as a loading control for Fig. 1C to demonstrate the specificity of the BACE1 band and to demonstrate that the reduction in sSEZ6 and sAPPb in the BACE1 KI line is indeed very similar to a full knock-out of BACE1. For all other BACE1 KO experiments the line BACE1^{tm1Psa} was used. The reason to use distinct BACE1 KO lines is due to the fact that this manuscript is a close collaboration between two different laboratories in different cities. While the Lichtenthaler group had always used the BACE1^{tm1Pcw} line in their previous studies, the Huth/Alzheimer group had always used the BACE1^{tm1Psa} line. Both KO lines constitute a full knock-out of BACE1 so that samples from either line would give the same result as a loading control.

In the methods section on page 13, we have now noted that the BACE1^{tm1Pcw} line was only used for Fig. 1C and its quantification in Fig. 1D.

2) The sex of the mice was not reported. For the behavioral experiments, sex differences need to be assessed.

The composition of each experimental group is now detailed in the Methods section. We stated that for the motor tests, no statistically significant differences were observed. For the EEG data, although a small number of sex-related differences reached statistical significance, this number was consistent with the false-positive rate expected by chance ($\alpha = 0.05$).

3) The authors should discuss that most of the phenotypes are common to BACE1-KO and BACE1-KI. The role of non-enzymatic BACE1 is minimal.

We agree with the reviewer and have added a sentence noting that most phenotypes are shared between BACE1-KO and BACE1-KI mice: *“Notwithstanding presumable effects of altered KCNQ-mediated currents on sleep cycle regulation, the behavioral and electroencephalographic differences between BACE1-KO and BACE1-WT were rather subtle. In contrast, the two transgenic mouse lines exhibited a number of common aberrations from the physiological sleep-wake cycles suggesting that it is predominantly the loss of enzymatic activity of BACE1 that defines the circadian phenotype of the mutant mice.”*

4) The authors should discuss that a limitation of the study is the use of germline KO and KI mice. Thus, these findings do not provide helpful information for eventual clinical trials with BACE inhibitors

We acknowledge that employing germline KO and KI models limits direct translation to clinical trials of BACE inhibitors initiated later in life: *“As we used germline KO and KI mice, we cannot make firm predictions on how a treatment regimen with BACE1- inhibitors beginning much later in life may affect sleep parameters, all the more so as patients are likely to be afflicted by disease-related sleep disorders. Despite the limited translational perspective of our study,...”*.

Referee #2:

Hannah Heining et al.'s study examined the effect of BACE1 on the sleep-wake cycle using BACE1 knockout (BACE1-KO) mice and a newly generated transgenic line that expresses a proteolysis-deficient variant of BACE1 (BACE1-KI). The researchers found that both BACE1-KO and BACE1-KI mice exhibited disturbances in their sleep-wake cycles. They proposed that BACE1 influences sleep architecture and regulates sleep-wake transitions through both proteolytic and non-proteolytic mechanisms. Consequently, the authors suggest that clinical trials involving BACE1-targeted inhibitors should take into account the impact of BACE1 on sleep-modulated behaviors. The findings are significant and relevant for a comprehensive evaluation of the balance between the therapeutic benefits and side effects of BACE1 inhibitors in recent clinical trials. Additionally, the study may provide valuable insights for broader interventions in sleep disorders.

However, certain concerns need to be addressed:

1) The authors should present a quantitative and statistical analysis of protein levels as depicted in Figure 1C.

As suggested by the reviewer, we quantified full-length APP to show the increase in full-length APP upon BACE1 knock-out/knock-in. Additionally, we quantified sSEZ6 and full-length SEZ6. The quantifications, which are now included as new Appendix Fig. S1, are in line with the representative blot shown in Fig. 1C.

2) In Figure 2E, the authors should also provide representative traces for BACE1-KO and BACE1-KI. Specifically, is there a difference in resting membrane potential among wild-type (WT), BACE1-KO, and BACE1-KI mice? The authors are also expected to describe their methodology for determining the results of XE991-induced membrane depolarization, as shown in Figure 1F, and explain how they concluded that M-currents were not altered. The reviewer recommends including a detailed description of these issues in the Materials and Methods section.

We thank the reviewer for this suggestion. Figure 2E has been revised to more clearly distinguish the different recordings; the corresponding values are now detailed in the figure legend, and the Methods section has been updated accordingly. No significant difference in resting membrane potential was observed across WT, KO, and KI mice (see revised Fig. 2E legend).

Referee #3:

The manuscript by Heininger et al reported the characterization of a knock-in mouse model with the mutation of the catalytic site in BACE1 (D289N). Authors in this study showed that this knock-in mice were unable to proteolytic process of BACE1 substrates, as evidenced in Figure 1. These results in Figure 1 by comparing that of BACE1 KO mice confirmed inactivation of proteolytic functions in BACE1 in this KI model. Authors further confirmed hypomyelination phenotypes which were reported to be seen in BACE1 KO mice. Importantly, authors identified clear differences of the hippocampal neuronal KCNQ-mediated M-current of the two mouse lines, in line with their previous finding that the non-enzymatic interaction between BACE1 and KCNQ in the control of M current. Authors in this study attempted to partially attribute this M-current to the observed differential sleep disturbance seen in KI vs KO mice. Nevertheless, this study reinforces the proteolytic functions in the regulation of BACE1 in the modulation of the thalamocortical networks such as sleep spindles in NREMS states. The manuscript is properly written and has a balanced discussion that supports the observations through comparisons of BACE1 KI and KO mice. While I am supportive to publication of this study, the minor concerns may need to be address.

1) Authors only did 24 hrs EEG recording and scored EEG in 12 hrs light phase and 11 hrs dark phase. Usually, it is expected to record mouse EEG for more than 24hrs, and score EEG in 12 hrs light phase and 12 hrs dark phase exclude the first and last 2hrs of the recording data to avoid the artificial interference of mice sleep-wake behavior. It is important to explain the logic of the experimental designs here.

We thank the referee for this comment and have clarified the EEG recordings. The previous text in the methods section

“Continuous recordings were performed using EGErA software for 24 h (with the final hour omitted for cage maintenance)” (Page 20. Last paragraph)

was changed to:

“Chronic EEG recordings were conducted for 23 hours per day over three consecutive days, consisting of 12 hours of light (Lights ON) and 11 hours of darkness (Lights OFF). The final hour of each dark phase was reserved for animal care (under red light), maintenance of the setup, and offline processing of the daily EEG data using the EGErA sleep scoring software (Cologne, Germany) (Fenzl et al., 2007; Touma et al., 2009; Romanowski et al., 2010; Fenzl et al., 2011; Polta et al., 2013; Kreuzer et al., 2015; Altunkaya et al., 2024).”

2) In figure 3 compressed EEG traces, NREM time (black spikes) in Bace1-ki control is much less than NREM time in BACE1 KI control during 11 hrs dark phase. Also, NREM time is visibly reduced in BACE1-KI compared to BACE1 KO during the dark phase. However, statistical analysis shows no difference between KI and KO mice (Figure 4A vs 4D). This discrepancy should be aware.

We are not entirely certain if we have interpreted the reviewer's comment correctly; nonetheless, to avoid possible misinterpretation, we replaced the representative BACE1-KI trace in Fig. 3 with another recording that more accurately illustrates the absence of a difference in NREMS duration, consistent with the statistical analysis shown in Fig. 4. Importantly, BACE1-KI and BACE1-KO were not directly compared statistically in this panel.

Dear Dr. Huth,

Thank you for the submission of your revised manuscript. It looks all good now, only a few editorial requests will need to be addressed before we can proceed with the official acceptance of your manuscript:

- Please provide up to 5 keywords with your ms file.
 - The conflict of interest statement needs to be renamed to "Disclosure and Competing Interests Statement" and placed after the Acknowledgments.
 - The author credits need to be removed from the ms file. All credits need to be entered during online ms submission.
 - The REFERENCE format needs to be corrected to the EMBO reports style: et al needs to be used after 10 author names; DOIs should only be used for preprints and datasets that have not been published yet. Please correct.
 - Suppl. Fig. 2 is not a correct callout, please correct.
 - The APPENDIX FILE needs page numbers in the table of content on the title page and "Supplementary Data" should be replaced by "Appendix".
 - "video" is mentioned in the text a few times, but no video files are uploaded.
 - The Methods section should include a Reagents and Tools Table (listing key reagents, experimental models, software and relevant equipment and including their sources and relevant identifiers) and a Methods and Protocols section in which the methods should be described using a step-by-step protocol format with bullet points, to facilitate the adoption of the methodologies across labs. More information on how to adhere to this format as well as downloadable templates (.docx) for the Reagents and Tools Table can be found in our author guidelines: <
<https://www.embopress.org/page/journal/14693178/authorguide#manuscriptpreparation>
 - Please label all panels of the source data clearly.
 - Materials and methods should be just Methods.
 - The specific URL for the PXD065011 dataset is not provided in the data availability statement, please add.
- * Figure Legends - Comments *
- Please define the annotated p values ****/**/*/* as well as provide the exact p-values for the same in the legend of figure S1 as appropriate.
 - Please note that the exact p values are not provided in the legends of figures 1D, E; 2D, F, J; 4A, C-G. Exact p-values need to be provided as reasonable.
 - Please note that information related to n is missing in the legend of figure 2F.
 - Please note that the error bars are not defined in the legend of figure S3.

I would like to suggest some minor changes to the abstract that needs to be written in present tense:

The β -secretase BACE1 has become a prime target in Alzheimer's disease (AD) therapy, because it drives the production of pathogenic amyloid β peptides. However, clinical trials with BACE1-targeting drugs were halted due to adverse effects on cognitive performance. We propose here that cognitive impairment by BACE1 inhibitors may be a corollary of a higher function of BACE1 related to proper sleep regulation. To address non-enzymatic effects of BACE1 on ion channels likely involved in the sleep-wake cycle, we analyze sleep patterns in both BACE1-KO mice and a newly generated transgenic line expressing a proteolysis-deficient BACE1 variant (BACE1-KI). We find that BACE1-KI and BACE1-KO mice display common and distinct sleep-wake disturbances. Compared with their respective wild-type littermates, both mutant lines sleep less during the light phase (when they preferentially rest). Furthermore, transition rates between wake and sleep states are altered, as are sleep spindles and EEG power spectra mainly in the gamma range. Thus, a better understanding of how BACE1 interferes with sleep-modulated behaviors is needed if clinical trials with BACE1-targeted inhibitors are to resume.

EMBO press papers are accompanied online by A) a short (1-2 sentences) summary of the findings and their significance, B) 2-3 bullet points highlighting key results and C) a synopsis image that is exactly 550 pixels wide and 200-600 pixels high (the height is variable). The synopsis image should provide a sketch of the major findings, like a graphical abstract. Please note that text needs to be readable at the final size. Please send us this information along with the final manuscript.

Dear Dr. Schnapp,

Thank you for the constructive revision process and the positive evaluation of our work. We have carefully addressed all editorial requests and updated the manuscript, figure legends, Appendix, Reagents and Tools Table, as well as the synopsis materials. We hope the submission now meets all requirements, and we remain happy to make any further adjustments if needed.

Kind regards,

Tobias Huth

- Please provide up to 5 keywords with your ms file.

Inserted 5 keywords after abstract.

- The conflict of interest statement needs to be renamed to "Disclosure and Competing Interests Statement" and placed after the Acknowledgments.

Changed

- The author credits need to be removed from the ms file. All credits need to be entered during online ms submission.

Removed

- The REFERENCE format needs to be corrected to the EMBO reports style: et al needs to be used after 10 author names; DOIs should only be used for preprints and datasets that have not been published yet. Please correct.

Corrected

- Suppl. Fig. 2 is not a correct callout, please correct.

Corrected to "Appendix Fig. S2"

- The APPENDIX FILE needs page numbers in the table of content on the title page and "Supplementary Data" should be replaced by "Appendix".

Added table of contents with page numbers and changed to "Appendix"

- "video" is mentioned in the text a few times, but no video files are uploaded.

The videos were deposited in the Zenodo repository. Added to Methods "Data have been deposited, see Data availability"

- The Methods section should include a Reagents and Tools Table (listing key reagents, experimental models, software and relevant equipment and including their sources and relevant identifiers) and a Methods and Protocols section in which the methods should be described using a step-by-step

protocol format with bullet points, to facilitate the adoption of the methodologies across labs. More information on how to adhere to this format as well as downloadable templates (.docx) for the Reagents and Tools Table can be found in our author guidelines: <

The table is now provided.

- Please label all panels of the source data clearly.

Thank you for the feedback. Could you please clarify what you mean by “label all panels of the source data”? Our Zenodo repository mirrors the organization used in recent EMBO Reports articles, and each dataset is referenced to the corresponding figure/panel in the manuscript.

- Materials and methods should be just Methods.

Changed

- The specific URL for the PXD065011 dataset is not provided in the data availability statement, please add.

The URL is now provided. The dataset will be published, once the manuscript is finally accepted.

* Figure Legends - Comments *

- Please define the annotated p values ****/***/**/* as well as provide the exact p-values for the same in the legend of figure S1 as appropriate.

“A-H” was added to figure 4 legend for the statistics part. p values are now stated in Appendix Figure S1.

- Please note that the exact p values are not provided in the legends of figures 1D, E; 2D, F, J; 4A, C-G. Exact p-values need to be provided as reasonable.

After consideration, we would like to keep the legends in threshold form (e.g., $p < 0.05$, $p < 0.01$, $p < 0.001$) rather than adding exact p-values in the figure legends. Exact P-values are given in the source data.

- Please note that information related to n is missing in the legend of figure 2F.

Thank you for pointing this out. The numbers were added to the figure legend.

- Please note that the error bars are not defined in the legend of figure S3.

Stated now in the legend.

I would like to suggest some minor changes to the abstract that needs to be written in present tense:

The β -secretase BACE1 has become a prime target in Alzheimer's disease (AD) therapy, because it

drives the production of pathogenic amyloid β peptides. However, clinical trials with BACE1-targeting drugs were halted due to adverse effects on cognitive performance. We propose here that cognitive impairment by BACE1 inhibitors may be a corollary of a higher function of BACE1 related to proper sleep regulation. To address non-enzymatic effects of BACE1 on ion channels likely involved in the sleep-wake cycle, we analyze sleep patterns in both BACE1-KO mice and a newly generated transgenic line expressing a proteolysis-deficient BACE1 variant (BACE1-KI). We find that BACE1-KI and BACE1-KO mice display common and distinct sleep-wake disturbances. Compared with their respective wild-type littermates, both mutant lines sleep less during the light phase (when they preferentially rest). Furthermore, transition rates between wake and sleep states are altered, as are sleep spindles and EEG power spectra mainly in the gamma range. Thus, a better understanding of how BACE1 interferes with sleep-modulated behaviors is needed if clinical trials with BACE1-targeted inhibitors are to resume.

Changed as requested.

EMBO press papers are accompanied online by A) a short (1-2 sentences) summary of the findings and their significance, B) 2-3 bullet points highlighting key results and C) a synopsis image that is exactly 550 pixels wide and 200-600 pixels high (the height is variable). The synopsis image should provide a sketch of the major findings, like a graphical abstract. Please note that text needs to be readable at the final size. Please send us this information along with the final manuscript.

Please find the requested Synopsis in the submission files.

Tobias Huth
Friedrich-Alexander-University Erlangen-Nürnberg, Germany
Institute of Physiology and Pathophysiology
Germany

Dear Dr. Huth,

I am very pleased to accept your manuscript for publication in the next available issue of EMBO reports. Thank you for your contribution to our journal.
